# Breaking the Linear Iteration Cost Barrier for Some Well-known Conditional Gradient Methods Using MaxIP Data-structures

**Zhaozhuo Xu**
Rice University
zx22@rice.edu

**Zhao Song**
Adobe Research
zsong@adobe.com

**Anshumali Shrivastava**
Rice University and ThirdAI Corp.
anshumali@rice.edu

## Abstract

Conditional gradient methods (CGM) are widely used in modern machine learning. CGM's overall running time usually consists of two parts: the number of iterations and the cost of each iteration. Most efforts focus on reducing the number of iterations as a means to reduce the overall running time. In this work, we focus on improving the per iteration cost of CGM. The bottleneck step in most CGM is maximum inner product search (MaxIP), which requires a linear scan over the parameters. In practice, approximate MaxIP data-structures are found to be helpful heuristics. However, theoretically, nothing is known about the combination of approximate MaxIP data-structures and CGM. In this work, we answer this question positively by providing a formal framework to combine the locality sensitive hashing type approximate MaxIP data-structures with CGM algorithms. As a result, we show the first algorithm, where the cost per iteration is sublinear in the number of parameters, for many fundamental optimization algorithms, e.g., Frank-Wolfe, Herding algorithm, and policy gradient.

## 1 Introduction

Conditional gradient methods (CGM), such as Frank-Wolfe and its variants, are well-known optimization approaches that have been extensively used in modern machine learning. For example, CGM has been applied to kernel methods [1, 2], structural learning [3] and online learning [4, 5, 6].

**Running Time Acceleration in Optimization:** Recent years have witnessed the success of large-scale machine learning models trained on vast amounts of data. In this learning paradigm, the computational overhead of most successful models is dominated by the optimization process [7, 8]. Therefore, reducing the running time of the optimization algorithm is of practical importance. The total running time in optimization can be decomposed into two components: (1) the number of iterations towards convergence, (2) the cost spent in each iteration. Reducing the number of iterations requires a better understanding of the geometric proprieties of the problem at hand and the invention of better potential functions to analyze the progress of the algorithm [9, 10, 11, 12, 13, 14, 15]. Reducing the cost spent per iteration usually boils down to designing problem-specific discrete data-structures. In the last few years, we have seen a remarkable growth of using data-structures to reduce iteration cost [16, 17, 18, 19, 20, 21, 22, 23, 24].

**MaxIP Data-structures for Iteration Cost Reduction:** A well-known strategy in optimization, with CGM, is to perform a greedy search over the weight vectors [9, 10, 13, 16, 25] or training

samples [26, 27] in each iteration. In this situation, the cost spent in each iteration is linear in the number of parameters. In practical machine learning, recent works [28, 29, 30, 31, 32] formulate this linear cost in iterative algorithms as an approximate maximum inner product search problem (MaxIP). They speed up the amortized cost per iteration via efficient data-structures from recent advances in approximate MaxIP [33, 34, 35, 36, 37, 38, 39, 40, 41, 42]. In approximate MaxIP data-structures, locality sensitive hashing (LSH) achieves promising performance with efficient random projection based preprocessing strategies [33, 34, 35, 36]. Such techniques are widely used in practice for cost reduction in optimization. [28] proposes an LSH based gradient sampling approach that reduces the total empirical running time of the adaptive gradient descent. [29] formulates the forward propagation of deep neural network as a MaxIP problem and uses LSH to select a subset of neurons for backpropagation. Therefore, the total running time of neural network training could be reduced to sublinear in the number of neurons. [31] extends this idea with system-level design for further acceleration, and [30] modifies the LSH with learning and achieves promising acceleration in attention-based language models. [32] formulates the greedy step in iterative machine teaching (IMT) as a MaxIP problem and scale IMT to large datasets with LSH.

**Challenges of Sublinear Iteration Cost CGM:** Despite the practical success of cost-efficient iterative algorithms with approximate MaxIP data-structure, the theoretical analysis of its combination with CGM is not well-understood. In this paper, we focus on this combination and target answering the following questions: (1) how to transform the iteration step of CGM algorithms into an approximate MaxIP problem? (2) how does the approximate error in MaxIP affect CGM in the total number of iterations towards convergence? (3) how to adapt approximate MaxIP data structure for iterative CGM algorithms?

**Our Contributions:** We propose a theoretical formulation for combining approximate MaxIP and convergence guarantees of CGM. In particular, we start with the popular Frank-Wolfe algorithm over the convex hull where the direction search in each iteration is a MaxIP problem. Next, we propose a sublinear iteration cost Frank-Wolfe algorithm using LSH type MaxIP data-structures. We then analyze the trade-off of approximate MaxIP and its effect on the number of iterations needed by CGM to converge. We show that the approximation error caused by LSH only leads to a constant multiplicative factor increase in the number of iterations. As a result, we retain the sub-linearly of LSH, with respect to the number of parameters, and at the same time retain the same asymptotic convergence as CGMs.

We summarize our complete contributions as follows.

- We give the first theoretical CGM formulation that achieves provable sublinear time cost per iteration. We also extend this result into Frank-Wolfe algorithm, Herding algorithm, and policy gradient method.
- We propose a pair of efficient transformations that formulates the direction search in Frank-Wolfe algorithm as a projected approximate MaxIP problem.
- We present the theoretical results that the proposed sublinear Frank-Wolfe algorithm asymptotically preserves the same order in the number of iterations towards convergence. Furthermore, we analyze the trade-offs between the saving in iteration cost and the increase in the number of iterations to accelerate total running time.
- We identify the problems of LSH type approximate MaxIP for cost reduction in the popular CGM methods and propose corresponding solutions.

The following sections are organized as below: Section 2 introduces the related works on data-structures and optimization, Section 3 introduces our algorithm associated with the main statements convergence, Section 4 provides the proof sketch of the main statements, Section 5 presents the societal impact and Section 6 concludes the paper.

## 2 Related work

### 2.1 Maximum Inner Product Search for Machine Learning

Maximum Inner Product Search (MaxIP) is a fundamental problem with applications in machine learning. Given a query $x \in \mathbb{R}^d$ and an $n$-vector dataset $Y \subset \mathbb{R}^d$, MaxIP targets at searching for $y \in Y$ that maximizes the inner product $x^\top y$. The naive MaxIP solution takes $O(dn)$ by comparing $x$ with each $y \in Y$. To accelerate this procedure, various algorithms are proposed to reduce the running time of MaxIP [33, 34, 36, 35, 37, 38, 43, 44, 39, 45, 40, 41, 42]. We could categorize

the MaxIP approaches into two categories: reduction methods and non-reduction methods. The reduction methods use transformations that transform approximate MaxIP into approximate nearest neighbor search (ANN) and solve it with ANN data-structures. One of the popular data-structure is locality sensitive hashing [46, 47].

**Definition 2.1** (Locality Sensitive Hashing). *Let $\overline{c} > 1$ denote a parameter. Let $p_1, p_2 \in (0, 1)$ denote two parameters and $p_1 > p_2$. We say a function family $\mathcal{H}$ is $(r, \overline{c} \cdot r, p_1, p_2)$-sensitive if and only if, for any vectors $x, y \in \mathbb{R}^d$, for any $h$ chosen uniformly at random from $\mathcal{H}$, we have:*

- *if $\|x - y\|_2 \leq r$, then $\mathrm{Pr}_{h \sim \mathcal{H}}[h(x) = h(y)] \geq p_1$,*

- *if $\|x - y\|_2 \geq \overline{c} \cdot r$, then $\mathrm{Pr}_{h \sim \mathcal{H}}[h(x) = h(y)] \leq p_2$.*

Here we define the LSH functions for euclidean distance. LSH functions could be used for search in cosine [48, 49] or Jaccard similarity [50, 51]. [33] first shows that MaxIP could be solved by $\ell_2$ LSH and asymmetric transformations. After that, [34, 36, 35, 43] propose a series of methods to solve MaxIP via LSH functions for other distance measures. Besides LSH, graph-based ANN approaches [38] could also be used after reduction.

On the other hand, the non-reduction method directly builds data-structures for approximate MaxIP. [37, 42] use quantization to approximate the inner product distance and build codebooks for efficient approximate MaxIP. [38, 44] propose a greedy algorithm for approximate MaxIP under computation budgets. [39, 40, 41] directly construct navigable graphs that achieve state-of-the-art empirical performance.

Recently, there is a remarkable growth in applying data-structures for machine learning [52, 53, 54]. Following the paradigm, approximate MaxIP data-structures have been applied to overcome the efficiency bottleneck of various machine learning algorithms. [38] formulates the inference of a neural network with a wide output layer as a MaxIP problem and uses a graph-based approach to reduce the inference time. In the same inference task, [55] proposes a learnable LSH data-structure that further improves the inference efficiency with less energy consumption. In neural network training, [29, 30, 31] uses approximate MaxIP to retrieve interested neurons for backpropagation. In this way, the computation overhead of gradient updates in neural networks could be reduced. In linear regression and classification models, [28] uses approximate MaxIP data-structures to retrieve the samples with large gradient norm and perform standard gradient descent, which improves the total running time for stochastic gradient descent. [32] proposes a scalable machine teaching algorithm that enables iterative teaching in large-scale datasets. In bandit problem, [56] proposes an LSH based algorithm that solves linear bandits problem with sublinear time complexity.

Despite the promising empirical results, there is little theoretical analysis on approximate MaxIP for machine learning. We summarize the major reasons as: (1) Besides LSH, the other approximate MaxIP data-structures do not provide theoretical guarantees on time and space complexity. (2) Current approaches treat data-structures and learning dynamics separately. There is no joint analysis on the effect of approximate MaxIP for machine learning.

## 2.2 Projection-free Optimization

Frank-Wolfe algorithm [25] is a projection-free optimization method with wide applications in convex [9, 10] and non-convex optimizations [11, 12]. The procedure of Frank-Wolfe algorithm could be summarized as two steps: (1) given the gradient, find a vector in the feasible domain that has the maximum inner product, (2) update the current weight with the retrieved vector. Formally, given a function $g : \mathbb{R}^d \rightarrow \mathbb{R}$ over a convex set $S$, starting from an initial weight $w^0$, the Frank-Wolfe algorithm updates the weight with learning rate $\eta$ follows:

$$s^t \leftarrow \arg\min_{s \in S} \langle s, \nabla g(w^t) \rangle$$

$$w^{t+1} \leftarrow (1 - \eta_t) \cdot w^t + \eta_t \cdot s^t.$$

Previous literature focuses on reducing the number of iterations for Frank-Wolfe algorithm over specific domains such as $\ell_p$ balls [9, 10, 13, 14]. The cost reduction in the iterative procedure of Frank-Wolfe algorithm is hardly discussed except [57]. In this work, we focus on the Frank-Wolfe algorithm over the convex hull of a finite feasible set. This formulation is more general and it includes recent Frank-Wolfe applications in probabilistic modeling [1, 2], structural learning [3] and policy optimization [5].

# 3 Our Sublinear Iteration Cost Algorithm

In this section, we formally present our results on the sublinear iteration cost CGM algorithms. We start with the preliminary definitions of the objective function. Then, we present the guarantees on the number of iteration and cost per iterations for our sublinear CGM algorithms to converge.

## 3.1 Preliminaries

We provide the notations and settings for this paper. We start with basic notations for this paper. For a positive integer $n$, we use $[n]$ to denote the set $\{1, 2, \cdots, n\}$. For a vector $x$, we use $\|x\|_2 := (\sum_{i=1}^n x_i^2)^{1/2}$ to denote its $\ell_2$ norm.

We say a function is convex if

$$L(x) \geq L(y) + \langle \nabla L(y), x - y \rangle.$$

We say a function is $\beta$-smooth if

$$L(y) \leq L(x) + \langle \nabla L(x), y - x \rangle + \frac{\beta}{2} \|y - x\|_2^2.$$

Given a set $A = \{x_i\}_{i \in [n]} \subset \mathbb{R}^d$, we say its convex hull $\mathcal{B}(A)$ is the collection of all finite linear combinations $y$ that satisfies $y = \sum_{i \in [n]} a_i \cdot x_i$, where $a_i \in [0, 1]$ for all $i \in [n]$ and $\sum_{i \in [n]} a_i = 1$. Let $D_{\max}$ denote the maximum diameter of $\mathcal{B}(A)$ so that $\|x - y\|_2 \leq D_{\max}$ for all $(x, y) \in \mathcal{B}(A)$. We present the detailed definitions in Appendix A.

Next, we present the settings of our work. Let $S \subset \mathbb{R}^D$ denote a $n$-point finite set. Given a convex and $\beta$-smooth function $g : \mathbb{R}^d \to \mathbb{R}$ defined over the convex hull $B(S)$, our goal is to find a $w \in B(S)$ that minimizes $g(w)$. Given large $n$ in the higher dimension, the dominant complexity of iteration cost lies in finding the MaxIP of $\nabla g(w)$ with respect to $S$. In this setting, the fast learning rate of Frank-Wolfe in $\ell_p$ balls [9, 13, 16] can not be achieved. We present the detailed problem setting of the Frank-Wolfe algorithm in Appendix C.

## 3.2 Our Results

We present our main results with comparison to the original algorithm in Table 2. From the table, we show that with near-linear preprocessing time, our algorithms maintain the same number of iterations towards convergence while reducing the cost spent in each iteration to be sublinear in the number of possible parameters.

| | Statement | Preprocess | #Iters | Cost per iter |
|---|---|---|---|---|
| Frank-Wolfe | [9] | 0 | $O(\beta D_{\max}^2/\epsilon)$ | $O(dn + \mathcal{T}_g)$ |
| Ours | Theorem 3.1 | $dn^{1+o(1)}$ | $O(\beta D_{\max}^2/\epsilon)$ | $O(dn^\rho + \mathcal{T}_g)$ |
| Herding | [1] | 0 | $O(D_{\max}^2/\epsilon)$ | $O(dn)$ |
| Ours | Theorem 3.2 | $dn^{1+o(1)}$ | $O(D_{\max}^2/\epsilon)$ | $O(dn^\rho)$ |
| Policy gradient | [5] | 0 | $O(\frac{\beta D_{\max}^2}{\epsilon^2(1-\gamma)^3\mu_{\min}^2})$ | $O(dn + \mathcal{T}_Q)$ |
| Ours | Theorem 3.3 | $dn^{1+o(1)}$ | $O(\frac{\beta D_{\max}^2}{\epsilon^2(1-\gamma)^3\mu_{\min}^2})$ | $O(dn^\rho + \mathcal{T}_Q)$ |

Table 1: Comparison between classical algorithm and our sublinear time algorithm. We compare our algorithm with Frank-Wolfe in: (1) "Frank-Wolfe" denotes Frank-Wolfe algorithm [9] for convex functions over a convex hull. Let $\mathcal{T}_g$ denote the time for evaluating the gradient for any parameter. (2) "Herding" denotes kernel Herding algorithm [1] (3) "Policy gradient" denotes the projection free policy gradient method [5]. Let $\mathcal{T}_Q$ denote the time for evaluating the policy gradient for any parameter. Let $\gamma \in (0, 1)$ denote the discount factor. Let $\mu_{\min}$ denote the minimum probability density of a state. Note that $n$ is the number of possible parameters. $n^{o(1)}$ is smaller than $n^c$ for any constant $c > 0$. Let $\rho \in (0, 1)$ denote a fixed parameter determined by LSH data-structure. The failure probability of our algorithm is $1/\operatorname{poly}(n)$. $\beta$ is the smoothness factor. $D_{\max}$ denotes the maximum diameter of the convex hull.

Next, we introduce the statement of our sublinear iteration cost algorithms. We start by introducing our result for improving the running time of Frank-Wolfe.

**Theorem 3.1** (Sublinear time Frank-Wolfe, informal of Theorem D.1). *Let $g : \mathbb{R}^d \to \mathbb{R}$ denote a convex and $\beta$-smooth function. Let the complexity of calculating $\nabla g(x)$ to be $\mathcal{T}_g$. Let $S \subset \mathbb{R}^d$ denote a set of $n$ points. Let $\mathcal{B} \subset \mathbb{R}^d$ denote the convex hull of $S$ with maximum diameter $D_{\max}$. Let $\rho \in (0,1)$ denote a fixed parameter. For any parameters $\epsilon, \delta$, there is an iterative algorithm (Algorithm 2) that takes $O(dn^{1+o(1)})$ time in pre-processing, takes $T = O(\beta D_{\max}^2/\epsilon)$ iterations and $O(dn^\rho + \mathcal{T}_g)$ cost per iteration, starts from a random $w^0$ from $\mathcal{B}$ as its initialization point,and outputs $w^T \in \mathbb{R}^d$ from $\mathcal{B}$ such that*

$$g(w^T) - \min_{w \in \mathcal{B}} g(w) \le \epsilon,$$

*holds with probability at least $1 - 1/\operatorname{poly}(n)$.*

Next, we show our main result for the Herding algorithm. Herding algorithm is widely applied in kernel methods [58]. [1] shows that the Herding algorithm is equivalent to a conditional gradient method with the least-squares loss function. Therefore, we extend our results and obtain the following statement.

**Theorem 3.2** (Sublinear time Herding algorithm, informal version of Theorem E.3). *Let $\mathcal{X} \subset \mathbb{R}^d$ denote a feature set and $\Phi : \mathbb{R}^d \to \mathbb{R}^k$ denote a mapping. Let $D_{\max}$ denote the maximum diameter of $\Phi(\mathcal{X})$ and $\mathcal{B}$ be the convex hull of $\Phi(\mathcal{X})$. Given a distribution $p(x)$ over $\mathcal{X}$, we denote $\mu = \mathbb{E}_{x \sim p(x)}[\Phi(x)]$. Let $\rho \in (0,1)$ denote a fixed parameter. For any parameters $\epsilon, \delta$, there is an iterative algorithm (Algorithm 3) that takes $O(dn^{1+o(1)})$ time in pre-processing, takes $T = O(D_{\max}^2/\epsilon)$ iterations and $O(dn^\rho)$ cost per iteration, starts from a random $w^0$ from $\mathcal{B}$ as its initialization point, and outputs $w^T \in \mathbb{R}^k$ from $\mathcal{B}$ such that*

$$\frac{1}{2}\|w^T - \mu\|_2^2 - \min_{w \in \mathcal{B}} \frac{1}{2}\|w - \mu\|_2^2 \le \epsilon,$$

*holds with probability at least $1 - 1/\operatorname{poly}(n)$.*

Finally, we present our result for policy gradient. Policy gradient [59] is a popular algorithm with wide applications in robotics [60] and recommendation [61]. [5] proposes a provable Frank-Wolfe method that maximizes the reward functions with policy gradient. However, the optimization requires a linear scan over all possible actions, which is unscalable in complex environments. We propose an efficient Frank-Wolfe algorithm with per iteration cost sublinear in the number of actions. Our statement is presented below.

**Theorem 3.3** (Sublinear time policy gradient, informal version of Theorem F.3). *Let $\mathcal{T}_Q$ denote the time for computing the policy graident. Let $D_{\max}$ denote the maximum diameter of action space and $\beta$ is a constant. Let $\gamma \in (0,1)$. Let $\rho \in (0,1)$ denote a fixed parameter. Let $\mu_{\min}$ denote the minimal density of states in $\mathcal{S}$. There is an iterative algorithm (Algorithm 5) that spends $O(dn^{1+o(1)})$ time in preprocessing, takes $O(\frac{\beta D_{\max}^2}{\epsilon^2(1-\gamma)^3\mu_{\min}^2})$ iterations and $O(dn^\rho + \mathcal{T}_Q)$ cost per iterations, starts from a random point $\pi_\theta^0$ as its initial point, and outputs $\pi_\theta^T$ that has the average gap $\sqrt{\sum_{s \in \mathcal{S}} g_T(s)^2} < \epsilon$ holds with probability at least $1 - 1/\operatorname{poly}(n)$, where $g_T(s)$ is defined in Eq. (6).*

## 4 Proof Overview

We present the overview of proofs in this section. We start with introducing the efficient MaxIP datastructures. Next, we show how to transform the direction search in a conditional gradient approach into a MaxIP problem. Finally, we provide proof sketches for each main statement in Section 3. The detailed proofs are presented in the supplement material.

### 4.1 Approximate MaxIP Data-structures

We present the LSH data-structures for approximate MaxIP in this section. The detailed description is presented in Appendix A. We use the reduction-based approximate MaxIP method with LSH data-structure to achieve sublinear iteration cost. Note that we choose this method due to its clear

theoretical guarantee on the retrieval results. It is well-known that an LSH data-structures is used for approximate nearest neighbor problem. The following definition of approximate nearest neighbor search is very standard in literature [62, 46, 47, 63, 64, 65, 66, 67, 68, 69, 70].

**Definition 4.1** (Approximate Nearest Neighbor (ANN)). *Let $\overline{c} > 1$ and $r \in (0, 2)$ denote two parameters. Given an $n$-vector set $Y \subset \mathbb{S}^{d-1}$ on a unit sphere, the objective of the $(\overline{c}, r)$-Approximate Nearest Neighbor (ANN) is to construct a data structure that, for any query $x \in \mathbb{S}^{d-1}$ such that $\min_{y \in Y} \|y - x\|_2 \leq r$, it returns a vector $z$ from $Y$ that satisfies $\|z - x\|_2 \leq \overline{c} \cdot r$.*

In the iterative-type optimization algorithm, the cost per iteration could be dominated by the Approximate MaxIP problem (Definition 4.2), which is the dual problem of the $(\overline{c}, r)$-ANN.

**Definition 4.2** (Approximate MaxIP). *Let $c \in (0, 1)$ and $\tau \in (0, 1)$ denote two parameters. Given an $n$-vector dataset $Y \subset \mathbb{S}^{d-1}$ on a unit sphere, the objective of the $(c, \tau)$-MaxIP is to construct a data structure that, given a query $x \in \mathbb{S}^{d-1}$ such that $\max_{y \in Y} \langle x, y \rangle \geq \tau$, it retrieves a vector $z$ from $Y$ that satisfies $\langle x, z \rangle \geq c \cdot \max_{y \in Y} \langle x, y \rangle$.*

Next, we present the the primal-dual connection between ANN and approximate MaxIP. Given to unit vectors $x, y \in \mathbb{R}^d$ with both norm equal to 1, $\|x - y\|_2^2 = 2 - 2\langle x, y \rangle$. Therefore, we could maximizing $\langle x, y \rangle$ by minimizing $\|x - y\|_2^2$. Based on this connection, we present how to solve $(c, \tau)$-MaxIP using $(\overline{c}, r)$-ANN. We start with showing how to solve $(\overline{c}, r)$-ANN with LSH.

**Theorem 4.3** (Andoni, Laarhoven, Razenshteyn and Waingarten [67]). *Let $\overline{c} > 1$ and $r \in (0, 2)$ denote two parameters. One can solve $(\overline{c}, r)$-ANN on a unit sphere in query time $O(d \cdot n^\rho)$ using preprocessing time $O(dn^{1+o(1)})$ and space $O(n^{1+o(1)} + dn)$, where $\rho = \frac{2}{\overline{c}^2} - \frac{1}{\overline{c}^4} + o(1)$.*

Next, we solve $(c, \tau)$-MaxIP by solving $(\overline{c}, r)$-ANN using Theorem 4.3. We have

**Corollary 4.4** (An informal statement of Corollary B.1). *Let $c \in (0, 1)$ and $\tau \in (0, 1)$ denote two parameters. One can solve $(c, \tau)$-MaxIP on a unit sphere $\mathcal{S}^{d-1}$ in query time $O(d \cdot n^\rho)$, where $\rho \in (0, 1)$, using LSH with both preprocessing time and space in $O(dn^{1+o(1)})$.*

In our work, we consider a generalized form of approximate MaxIP, denoted as projected approximate MaxIP.

**Definition 4.5** (Projected approximate MaxIP). *Let $\phi, \psi : \mathbb{R}^d \to \mathbb{R}^k$ denote two transforms. Given an $n$-vector dataset $Y \subset \mathbb{R}^d$ so that $\psi(Y) \subset \mathbb{S}^{k-1}$, the goal of the $(c, \phi, \psi, \tau)$-MaxIP is to construct a data structure that, given a query $x \in \mathbb{R}^d$ and $\phi(x) \in \mathbb{S}^{k-1}$ such that $\max_{y \in Y} \langle \phi(x), \psi(y) \rangle \geq \tau$, it retrieves a vector $z \in Y$ that satisfies $\langle \phi(x), \psi(z) \rangle \geq c \cdot (\phi, \psi)$-MaxIP$(x, Y)$.*

For details of space-time trade-offs, please refer to Appendix B. The following sections show how to use projected approximate MaxIP to accelerate the optimization algorithm by reducing the cost per iteration.

## 4.2 Efficient Transformations

We have learned from Section 4.1 that $(c, \tau)$-MaxIP on a unit sphere $\mathcal{S}^{d-1}$ using LSH for ANN. Therefore, the next step is to transform the direction search procedure in iterative optimization algorithm into a MaxIP on a unit sphere. To achieve this, we formulate the direction search as a projected approximate MaxIP (see Definition A.5). We start with presenting a pair of transformation $\phi_0, \psi_0 : \mathbb{R}^d \to \mathbb{R}^{d+1}$ such that, given a function $g : \mathbb{R}^d \to \mathbb{R}$, for any $x, y$ in a convex set $\mathcal{K}$, we have

$$\phi_0(x) := [\nabla g(x)^\top, x^\top \nabla g(x)]^\top, \quad \psi_0(y) := [-y^\top, 1]^\top. \tag{1}$$

In this way, we show that

$$\langle y - x, \nabla g(x) \rangle = -\langle \phi_0(x), \psi_0(y) \rangle,$$
$$\arg\min_{y \in Y} \langle y - x, \nabla g(x) \rangle = \arg\max_{y \in Y} \langle \phi_0(x), \psi_0(y) \rangle \tag{2}$$

Therefore, we could transform the direction search problem into a MaxIP problem.

Next, we present a standard transformations [36] that connects the MaxIP to ANN in unit sphere. For any $x, y \in \mathbb{R}^d$, we propose transformation $\phi_1, \psi_1 : \mathbb{R}^d \to \mathbb{R}^{d+2}$ such that

$$\phi_1(x) = \begin{bmatrix} (D_x^{-1}x)^\top & 0 & \sqrt{1 - \|xD_x^{-1}\|_2^2} \end{bmatrix}^\top$$

$$\psi_1(y) = \begin{bmatrix} (D_y^{-1}y)^\top & \sqrt{1 - \|yD_y^{-1}\|_2^2} & 0 \end{bmatrix}^\top \tag{3}$$

Here $D_x$, $D_y$ are some constant that make sure both $x/D_x$ and $y/D_y$ have norms less than 1. Under these transformations, both $\phi_1(x)$ and $\psi_1(y)$ have norm 1 and $\arg\max_{y \in Y}\langle \phi_1(x), \psi_1(y)\rangle = \arg\max_{y \in Y}\langle x, y\rangle$.

Combining transformations in Eq. (1) and Eq. (3), we obtain query transform $\phi : \mathbb{R}^d \to \mathbb{R}^{d+3}$ with form $\phi(x) = \phi_1(\phi_0(x))$ and data transform $\phi : \mathbb{R}^d \to \mathbb{R}^{d+3}$ with form $\psi(y) = \psi_1(\psi_0(y))$. Using $\phi$ and $\psi$, we transform the direction search problem in optimization into a MaxIP in unit sphere. Moreover, given a set $Y \subset \mathbb{R}^d$ and a query $x \in \mathbb{R}^d$, the solution $z$ of $(c, \phi, \psi, \tau)$-MaxIP over $(x, Y)$ has the propriety that $\langle z - x, \nabla g(x)\rangle \leq c \cdot \min_{y \in Y}\langle y - x, \nabla g(x)\rangle$. Thus, we could approximate the direction search with LSH based MaxIP data-structure.

Note that only MaxIP problem with positive inner product values could be solved by LSH. We found the direction search problem naturally satisfies this condition. We show that if $g$ is convex, given a set $S \subset \mathbb{R}^d$, we have $\min_{s \in S}\langle \nabla g(x), s - x\rangle \leq 0$ for any $x \in \mathcal{B}(S)$, where $\mathcal{B}$ is the convex hull of $S$. Thus, $\max_{y \in Y}\langle \phi_0(x), \psi_0(y)\rangle$ is non-negative following Eq. (2).

### 4.3 Proof of Theorem 3.1

We present the proof sketch for Theorem 3.1 in this section. We refer the readers to Appendix D for the detailed proofs.

Let $g : \mathbb{R}^d \to \mathbb{R}$ denote a convex and $\beta$-smooth function. Let the complexity of calculating $\nabla g(x)$ to be $\mathcal{T}_g$. Let $S \subset \mathbb{R}^d$ denote a set of $n$ points, and $\mathcal{B} \subset \mathbb{R}^d$ be the convex hull of $S$ with maximum diameter $D_{\max}$. Let $\phi, \psi : \mathbb{R}^d \to \mathbb{R}^{d+3}$ denote the tranformations defined in Section 4.2. Starting from a random vector $w^0 \in \mathcal{B}(S)$. Our sublinear Frank-Wolfe algorithm follows the update following rule that each step

$$s^t \leftarrow (c, \phi, \psi, \tau)\text{-MaxIP of } w^t \text{ with respect to } S$$
$$w^{t+1} \leftarrow w^t + \eta \cdot (s^t - w^t)$$

We start with the upper bounding $\langle s^t - w^t, \nabla g(w^t)\rangle$. Because $s^t$ is the $(c, \phi, \psi, \tau)$-MaxIP of $w^t$ with respect to $S$, we have

$$\langle s^t - w^t, \nabla g(w^t)\rangle \leq c \min_{s \in S}\langle s - w^t, \nabla g(w^t)\rangle \leq c\langle w^* - w^t, \nabla g(w^t)\rangle \tag{4}$$

For convenient of the proof, for each $t$, we define $h_t = g(w^t) - g(w^*)$. Next, we upper bound $h_{t+1}$ as

$$h_{t+1} \leq g(w^t) + \eta_t\langle s^t - w^t, \nabla g(w^t)\rangle + \frac{\beta}{2}\eta_t^2\|s^t - w^t\|_2^2 - g(w^*)$$

$$\leq g(w^t) + c\eta_t\langle w^* - w^t, \nabla g(w^t)\rangle + \frac{\beta}{2}\eta_t^2\|s^t - w^t\|_2^2 - g(w^*)$$

$$\leq g(w^t) + c\eta_t\langle w^* - w^t, \nabla g(w^t)\rangle + \frac{\beta D_{\max}^2}{2}\eta_t^2 - g(w^*)$$

$$\leq (1 - \eta_t)g(w^t) + c\eta_t g(w^*) + \frac{\beta D_{\max}^2}{2}\eta_t^2 - g(w^*)$$

$$= (1 - c\eta_t)h_t + \frac{\beta D_{\max}^2}{2}\eta_t^2$$

$$\tag{5}$$

where the first step follows from the definition of $\beta$-smoothness, the second step follows from Eq. (4), the third step follows from the definition of $D_{\max}$, the forth step follows from the convexity of $g$.

Let $\eta = \frac{2}{c(t+2)}$ and $A_t = \frac{t(t+1)}{2}$. Combining them with Eq.(5), we show that

$$A_{t+1}h_{t+1} - A_t h_t = c^{-2}\frac{t+1}{t+2}\beta D_{\max}^2$$
$$< c^{-2}\beta D_{\max}^2$$

Using induction from 1 to $t$, we show that

$$A_t h_t < c^{-2}t\beta D_{\max}^2$$

Taken $A_t = \frac{t(t+1)}{2}$ into consideration, we have

$$h_t < \frac{2\beta D_{\max}^2}{c^2(t+1)}$$

Given constant approximation ratio $c$, $t$ should be in $O(\frac{\beta D_{\max}^2}{\epsilon})$ so that $h_t \le \epsilon$. Thus, we complete the proof.

**Cost Per Iteration** After we take $O(dn^{1+o(1)})$ preprocessing time, the cost per iteration consists three pairs: (1) it takes $\mathcal{T}_g$ to compute $\nabla g(w^t)$, (2) it takes $O(d)$ to perform transform $\phi$ and $\psi$, (3) it takes $O(dn^\rho)$ to retrieve $s^t$ from LSH. Thus, the final cost per iteration would be $O(dn^\rho + \mathcal{T}_g)$.

Next, we show how to extend the proof to Herding problem. Following [1], we start with defining function $g = \frac{1}{2}\|w^T - \mu\|_2^2$. We show that this function $g$ is a convex and 1-smooth function. Therefore, the Herding algorithm is equivalent to the Frank-Wolfe Algorithm over function $g$. Using the proof of Theorem 3.1 with $\beta = 1$, we show that it takes $T = O(D_{\max}^2/\epsilon)$ iterations and $O(dn^\rho)$ cost per iteration to reach the $\epsilon$-optimal solution. Similar to Theorem 3.1, we show that the cost per iteration would be $O(dn^\rho)$ as it takes $O(d)$ to compute $\nabla g(w^t)$.

### 4.4 Proof of Theorem 3.3

We present the proof sketch for Theorem 3.3 in this section. We refer the readers to Appendix F for the detailed proofs.

In this paper, we focus on the action-constrained Markov Decision Process (ACMDP). In this setting, we are provided with a state $\mathcal{S} \in \mathbb{R}^k$ and action space $\mathcal{A} \in \mathbb{R}^d$. However, at each step $t \in \mathbb{N}$, we could only access a finite $n$-vector set of actions $\mathcal{C}(s) \subset \mathcal{A}$. Let us assume the $\mathcal{C}(s)$ remains the same in each step. Let us denote $D_{\max}$ as the maximum diameter of $\mathcal{A}$.

When you play with this ACMDP, the policy you choose is defined as $\pi_\theta(s) : \mathcal{S} \to \mathcal{A}$ with parameter $\theta$. Meanwhile, there exists a reward function $r : \mathcal{S} \times \mathcal{A} \in [0, 1]$. Then, we define the Q function as below,

$$Q(s, a|\pi_\theta) = \mathbb{E}\Big[ \sum_{t=0}^{\infty} \gamma^t r(s_t, a_t)|s_0 = s, a_0 = a, \pi_\theta \Big].$$

where $\gamma \in (0, 1)$ is a discount factor.

Given a state distribution $\mu$, the objective of policy gradient is to maximize $J(\mu, \pi_\theta) = \mathbb{E}_{s\sim\mu, a\sim\pi_\theta}[Q(s, a|\pi_\theta)]$ via policy gradient [59] denoted as:

$$\nabla_\theta J(\mu, \pi_\theta) = \mathbb{E}_{s \sim d_\mu^\pi} \Big[ \nabla_\theta \pi_\theta(s)\nabla_a Q(s, \pi_\theta(s)|\pi_\theta)| \Big].$$

[5] propose an iterative algorithm that perform MaxIP at each iteration $k$ over actions to find

$$g_k(s) = \max_{a \in \mathcal{C}(s)} \langle a_s^k - \pi_\theta^k(s), \nabla_a Q(s, \pi_\theta^k(s)|\pi_\theta^k))\rangle. \qquad (6)$$

In this work, we approximate Eq. (6) using $(c, \phi, \psi, \tau)$-MaxIP. Here define $\phi : \mathcal{S} \times \mathbb{R}^d \to \mathbb{R}^{d+1}$ and $\psi : \mathbb{R}^d \to \mathbb{R}^{d+1}$ as follows:

$$\phi(s, \pi_\theta^k) := [\nabla_a Q(s, \pi_\theta^k(s)|\pi_\theta^k)^\top, (\pi_\theta^k)^\top Q(s, \pi_\theta^k(s)|\pi_\theta^k)]^\top, \psi(a) := [a^\top, -1]^\top.$$

Then, we have $g_k(s) = \langle \phi(s, \pi_\theta^k), \psi(a) \rangle$. Note that we still require transformations in Eq. (3) to generate unit vectors.

Next, we show that if we retrieve an action $\widehat{a_s^k}$ using $(c, \phi, \psi, \tau)$-MaxIP, the gap $\widehat{g}_k(s)$ would be lower bounded by

$$\widehat{g}_k(s) = \langle \widehat{a_s^k} - \pi_\theta^k(s), \nabla_a Q(s, \pi_\theta^k(s)|\pi_\theta^k)) \rangle$$
$$\geq c g_k(s) \tag{7}$$

Combining Eq. (7) the standard induction in [5], we upper bound $\sum_{s \in \mathcal{S}} g_T(s)^2$ as

$$\sum_{s \in \mathcal{S}} g_T(s)^2 \leq \frac{1}{T+1} \frac{2\beta D_{\max}^2}{c^2(1-\gamma)^3 \mu_{\min}^2}. \tag{8}$$

where $\mu_{\min}$ denotes the minimal density of sates in $\mathcal{S}$ and $\beta$ is the smoothness factor.

In this way, given a constant factor $c$, if we would like to minimize the gap $\sum_{s \in \mathcal{S}} g_T(s)^2 < \epsilon^2$, $T$ should be $O(\frac{\beta D_{\max}^2}{\epsilon^2(1-\gamma)^3 \mu_{\min}^2})$.

**Cost Per Iteration** After we take $O(dn^{1+o(1)})$ preprocessing time, the cost per iteration consists three pairs: (1) it takes $\mathcal{T}_Q$ to compute policy gradient, (2) it takes $O(d)$ to perform transform $\phi$ and $\psi$, (3) it takes $O(dn^\rho)$ to retrieve actions from LSH. Thus, the final cost per iteration would be $O(dn^\rho + \mathcal{T}_Q)$.

### 4.5 Quantization for Adaptive Queries

In optimization, the gradient computed in every iteration is not independent of each other. This would generate a problem for MaxIP data-structures. If we use a vector containing the gradients as a query for MaxIP data-structures, the query failure probability in each iteration is not independent. Therefore, the total failure probability could not be union bounded. As previous MaxIP data-structures focus on the assumptions that queries are independent, the original failure analysis could not be directly applied.

This work uses a standard query quantization method to handle the adaptive query sequence in optimization. Given the known query space, we quantize it by lattices [71]. This quantization is close to the Voronoi diagrams. In this way, each query is located into a cell with a center vector. Next, we perform a query using the center vector in the cell. Therefore, the failure probability of the MaxIP query sequence is equivalent to the probability that any center vector in the cell fails to retrieve its approximate MaxIP solution. As the centers of cells are independent, we could union bound this probability. On the other hand, as the maximum diameter of the cell is $\lambda$, this query quantization would introduce a $\lambda$ additive error in the inner product retrieved. We refer the readers to Appendix G for the detailed quantization approach.

### 4.6 Optimizing Accuracy-Efficiency Trade-off

In this work, we show that by LSH based MaxIP data-structure, the cost for direction search is $O(dn^\rho)$, where $\rho \in (0, 1)$. In Section D.2 of the supplementary material, we show that $\rho$ is a function of constant $c$ and parameter $\tau$ in approximate MaxIP (see Definition 4.2). Moreover, we also show in Section D.2 that LSH results in only a constant multiplicative factor increase in the number of iterations. Considering the cost per iteration and the number of iterations, we show that when our algorithms stop at the $\epsilon$-optimal solution, LSH could achieve acceleration in the overall running time. Therefore, we could set $c$ and $\tau$ parameters to balance the accuracy-efficiency trade-off of CGM to achieve the desired running time.

## 5 Potential Negative Societal Impact

This paper discusses the theoretical foundation of data-structures for conditional gradient methods. We believe that this paper does not have negative societal impact in the environment, privacy, and other domains.

# 6 Concluding Remarks

In this work, we present the first Frank-Wolfe algorithms that achieve sublinear linear time cost per iteration. We also extend this result into Herding algorithm and policy gradient methods. We formulate the direction search in Frank-Wolfe algorithm as a projected approximate maximum inner product search problem with a pair of efficient transformations. Then, we use locality sensitive hashing data-structure to reduce the iteration cost into sublinear over the number of possible parameters. Our theoretical analysis shows that the sublinear iteration cost Frank-Wolfe algorithm preserves the same order in the number of iterations towards convergence. Moreover, we analyze and optimize the trade-offs between saving iteration cost and increasing the number of iterations to achieve sublinear total running time. Furthermore, we identify the problems of existing MaxIP data-structures for cost reduction in iterative optimization algorithms and propose the corresponding solutions. We hope this work can be the starting point of future study on sublinear iteration cost algorithms for optimization.

## Acknowledgements

This work was supported by National Science Foundation IIS-1652131, BIGDATA-1838177, AFOSR-YIP FA9550-18-1-0152, ONR DURIP Grant, and the ONR BRC grant on Randomized Numerical Linear Algebra. The authors would like to thank Beidi Chen for the helpful discussion on optimization. The authors would like to thank Lichen Zhang for valuable discussions about data-structures.

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
