# Appendix

**Roadmap.** We provide supplementary materials for our work. Section A introduces the preliminary notations and definitions, Section B introduces the LSH data structure in detail for MaxIP, Section C presents our sublinear Frank-Wolfe algorithm, Section D presents the convergence analysis for sublinear Frank-Wolfe, Section E provides the algorithm and analysis on sublinear cost Herding algorithm, Section F provides the algorithm and analysis on sublinear cost policy gradient approach, Section G shows how to handle adaptive queries in MaxIP.

## A Preliminary

### A.1 Notations

We use $\Pr[]$ and $\mathbb{E}[]$ for probability and expectation. We denote $\max\{a, b\}$ as the maximum between $a$ and $b$. We denote $\min\{a, b\}$ (resp. $\max\{a, b\}$) as the minimum (reps. maximum) between $a$ and $b$. For a vector $x$, we denote $\|x\|_2 := (\sum_{i=1}^n x_i^2)^{1/2}$ as its $\ell_2$ norm. We denote $\|x\|_p := (\sum_{i=1}^n |x_i|^p)^{1/p}$ as its $\ell_p$ norm. For a square matrix $A$, we denote $\mathrm{tr}[A]$ as the trace of matrix $A$.

### A.2 LSH and MaxIP

We start with the defining the Approximate Nearest Neighbor (ANN) problem [62, 46, 47, 63, 64, 65, 66, 67, 68, 69, 70] as:

**Definition A.1** (Approximate Nearest Neighbor (ANN)). *Let $\bar{c} > 1$ and $r \in (0, 2)$ denote two parameters. Given an $n$-vector set $Y \subset \mathbb{S}^{d-1}$ on a unit sphere, the objective of the $(\bar{c}, r)$-Approximate Nearest Neighbor (*ANN*) is to construct a data structure that, for any query $x \in \mathbb{S}^{d-1}$ such that $\min_{y \in Y} \|y - x\|_2 \leq r$, it returns a vector $z$ from $Y$ that satisfies $\|z - x\|_2 \leq \bar{c} \cdot r$.*

The ANN problem can be solved via locality sensitive hashing (LSH) [46, 47, 66]. In this paper, we use the standard definitions of LSH (see Indyk and Motwani [46]).

**Definition A.2** (Locality Sensitive Hashing). *Let $\bar{c} > 1$ denote a parameter. Let $p_1, p_2 \in (0, 1)$ denote two parameters and $p_1 > p_2$. We say a function family $\mathcal{H}$ is $(r, \bar{c} \cdot r, p_1, p_2)$-sensitive if and only if, for any vectors $x, y \in \mathbb{R}^d$, for any $h$ chosen uniformly at random from $\mathcal{H}$, we have:*

- *if $\|x - y\|_2 \leq r$, then $\Pr_{h \sim \mathcal{H}}[h(x) = h(y)] \geq p_1$,*
- *if $\|x - y\|_2 \geq \bar{c} \cdot r$, then $\Pr_{h \sim \mathcal{H}}[h(x) = h(y)] \leq p_2$.*

Next, we show that LSH solves ANN problem with sublinear query time complexity.

**Theorem A.3** (Andoni, Laarhoven, Razenshteyn and Waingarten [67]). *Let $\bar{c} > 1$ and $r \in (0, 2)$ denote two parameters. One can solve $(\bar{c}, r)$-ANN on a unit sphere in query time $O(d \cdot n^\rho)$ using preprocessing time $O(dn^{1+o(1)})$ and space $O(n^{1+o(1)} + dn)$, where $\rho = \frac{2}{\bar{c}^2} - \frac{1}{\bar{c}^4} + o(1)$.*

Here we write $o(1)$ is equivalent to $O(1/\sqrt{\log n})$. Note that we could reduce $d$ to $n^{o(1)}$ with Johnson–Lindenstrauss Lemma [72]. Besides, we could achieve better $\rho$ using LSH in [65] if we allowed to have more proprocessing time.

In this work, we focus on a well-known problem in computational complexity: approximate MaxIP. In this work, we follow the standard notation in [73] and define the approximate MaxIP problem as follows:

**Definition A.4** (Approximate MaxIP). *Let $c \in (0, 1)$ and $\tau \in (0, 1)$ denote two parameters. Given an $n$-vector dataset $Y \subset \mathbb{S}^{d-1}$ on a unit sphere, the objective of the $(c, \tau)$-MaxIP is to construct a data structure that, given a query $x \in \mathbb{S}^{d-1}$ such that $\max_{y \in Y} \langle x, y \rangle \geq \tau$, it retrieves a vector $z$ from $Y$ that satisfies $\langle x, z \rangle \geq c \cdot \max_{y \in Y} \langle x, y \rangle$.*

In many applications, it is more convenient to doing inner product search in a transformed/projected space compared to doing inner product search in the original space. Thus, we propose the following definitions (Definition A.5 and Definition A.6)

**Definition A.5** (Projected MaxIP). *Let $\phi, \psi : \mathbb{R}^d \to \mathbb{R}^k$ denote two transforms. Given a data set $Y \subseteq \mathbb{R}^d$ and a point $x \in \mathbb{R}^d$, we define $(\phi, \psi)$-MaxIP as follows:*

$$(\phi, \psi)\text{-MaxIP}(x, Y) := \max_{y \in Y} \langle \phi(x), \psi(y) \rangle$$

**Definition A.6** (Projected approximate MaxIP). *Let $\phi, \psi : \mathbb{R}^d \to \mathbb{R}^k$ denote two transforms. Given an $n$-vector dataset $Y \subset \mathbb{R}^d$ so that $\psi(Y) \subset \mathbb{S}^{k-1}$, the goal of the $(c, \phi, \psi, \tau)$-MaxIP is to construct a data structure that, given a query $x \in \mathbb{R}^d$ and $\phi(x) \in \mathbb{S}^{k-1}$ such that $\max_{y \in Y} \langle \phi(x), \psi(y) \rangle \geq \tau$, it retrieves a vector $z \in Y$ that satisfies $\langle \phi(x), \psi(z) \rangle \geq c \cdot (\phi, \psi)$-MaxIP$(x, Y)$.*

Besides MaxIP, We also define a version of the minimum inner product search problem.

**Definition A.7** (Regularized Min-IP). *Given a data set $Y \subseteq \mathbb{R}^d$ and a point $x \in \mathbb{R}^d$. Let $\phi : \mathbb{R}^d \to \mathbb{R}^d$ denote a mapping. Given a constant $\alpha$, we define regularized Min-IP as follows:*

$$(\phi, \alpha)\text{-Min-IP}(x, Y) := \min_{y \in Y} \langle y - x, \phi(x) \rangle + \alpha \|x - y\|.$$

### A.3 Definitions and Properties for Optimization

We start with listing definitions for optimization.

**Definition A.8** (Convex hull and its diameter). *Given a set $A = \{x_i\}_{i \in [n]} \subset \mathbb{R}^d$, we define its convex hull $\mathcal{B}(A)$ to be the collection of all finite linear combinations $y$ that satisfies $y = \sum_{i \in [n]} a_i \cdot x_i$ where $a_i \in [0, 1]$ for all $i \in [n]$ and $\sum_{i \in [n]} a_i = 1$. Let $D_{\max}$ denote the maximum square of diameter of $\mathcal{B}(A)$ so that $\|x - y\|_2 \leq D_{\max}$ for all $(x, y) \in \mathcal{B}(A)$.*

**Definition A.9** (Smoothness). *We say $L$ is $\beta$-smooth if*

$$L(y) \leq L(x) + \langle \nabla L(x), y - x \rangle + \frac{\beta}{2} \|y - x\|_2^2$$

**Definition A.10** (Convex). *We say function $L$ is convex if*

$$L(x) \geq L(y) + \langle \nabla L(y), x - y \rangle$$

Next, we list properties for optimization.

**Corollary A.11.** *For a set $A = \{x_i\}_{i \in [n]} \subset \mathbb{R}^d$, and its convex hull $\mathcal{B}(A)$, given a query $q \in \mathbb{R}^d$, if $x^* = \arg \max_{x \in A} q^\top x$. Then, $q^\top y \leq q^\top x^*$ for all $y \in \mathcal{B}(A)$.*

*Proof.* We can upper bound $q^\top y$ as follows:

$$
\begin{aligned}
q^\top y &= q^\top \left( \sum_{i \in [n]} a_i \cdot x_i \right) \\
&= \sum_{i \in [n]} a_i \cdot q^\top x_i \\
&\leq \sum_{i \in [n]} a_i \cdot q^\top x^* \\
&\leq q^\top x^*
\end{aligned}
$$

where the first step follows from the definition of convex hull in Definition A.8, the second step is an reorganization, the third step follows the fact that $a_i \in [0, 1]$ for all $i \in [n]$ and $q^\top x_i \leq q^\top x^*$ for all $x_i \in A$, the last step follows that $\sum_{i \in [n]} a_i = 1$. $\qquad \square$

**Lemma A.12** (MaxIP condition). *Let $g : \mathbb{R}^d \to \mathbb{R}$ denote a convex function. Let $S \subset \mathbb{R}^d$ denote a set of points. Given a vector $x \in \mathcal{B}(S)$, we have*

$$\min_{s \in S} \langle \nabla g(x), s - x \rangle \leq 0, \quad \forall x \in \mathcal{B}(\mathcal{S}).$$

*Proof.* Let $s_{\min} = \arg\min_{s \in S} \langle \nabla g(x), s \rangle$. Then, we upper bound $\langle \nabla g(x), s_{\min} - x \rangle$ as

$$\langle \nabla g(x), s_{\min} - x \rangle = \langle \nabla g(x), s_{\min} - \sum_{s \in S} a_i \cdot s \rangle$$

$$\leq \langle \nabla g(x), \sum_{s \in S} a_i(s_{\min} - s_i) \rangle$$

$$= \sum_{s_i \in S} a_i \langle \nabla g(x), s_{\min} - s_i \rangle$$

$$= \sum_{s_i \in S} a_i(\langle \nabla g(x), s_{\min} \rangle - \langle \nabla g(x), s_i \rangle)$$

$$\leq 0 \tag{9}$$

where the first step follows from the definition of convex hull in Definition A.8, the second and third steps are reorganizations, the final steps follows that $\langle \nabla g(x), s_0 \rangle \leq \langle \nabla g(x), s \rangle$ for all $s \in S$.

Next, we upper bound $\min_{s \in S} \langle \nabla g(x), s - x \rangle \leq 0, \quad \forall x \in \mathcal{B}(\mathcal{S})$ as

$$\min_{s \in S} \langle \nabla g(x), s - x \rangle \leq \langle \nabla g(x), s_0 - x \rangle \leq 0$$

where the first step follows from the definition of function $\min$ and the second step follows from Eq (9). $\qquad \square$

# B   Data Structures

In this section, we present a formal statement that solves $(c, \tau)$-MaxIP problem on unit sphere using LSH for $(\bar{c}, r)$-ANN.

**Corollary B.1** (Formal statement of Corollary 4.4)**.** *Let $c \in (0, 1)$ and $\tau \in (0, 1)$. Given a set of $n$-vector set $Y \subset \mathcal{S}^{d-1}$ on the unit sphere, there exists a data structure with $O(dn^{1+o(1)})$ preprocessing time and $O(n^{1+o(1)} + dn)$ space so that for any query $x \in \mathcal{S}^{d-1}$, we take $O(d \cdot n^\rho)$ query time to retrieve the $(c, \tau)$-MaxIP of $x$ in $Y$ with probability at least $0.9$[i], where $\rho := \frac{2(1-\tau)^2}{(1-c\tau)^2} - \frac{(1-\tau)^4}{(1-c\tau)^4} + o(1)$*

*Proof.* We know that $\|x - y\|_2^2 = 2 - 2\langle x, y \rangle$ for all $x, y \in \mathcal{S}^{d-1}$. In this way, if we have a LSH data-structure for $(\bar{c}, r)$-ANN. It could be used to solve $(c, \tau)$-MaxIP with $\tau = 1 - 0.5r^2$ and $c = \frac{1 - 0.5\bar{c}^2 r^2}{1 - 0.5r^2}$. Next, we write $\bar{c}^2$ as

$$\bar{c}^2 = \frac{1 - c(1 - 0.5r^2)}{0.5r^2} = \frac{1 - c\tau}{1 - \tau}.$$

Next, we show that if the LSH is initialized following Theorem A.3, it takes query time $O(d \cdot n^\rho)$, space $O(n^{1+o(1)} + dn)$ and preprocessing time $O(dn^{1+o(1)})$ to solve $(c, \tau)$-MaxIP through solving $(\bar{c}, r)$-ANN, where

$$\rho = \frac{2}{\bar{c}^2} - \frac{1}{\bar{c}^4} + o(1) = \frac{2(1-\tau)^2}{(1-c\tau)^2} - \frac{(1-\tau)^4}{(1-c\tau)^4} + o(1).$$

$\qquad \square$

In practice, $c$ is increasing as we set parameter $\tau$ close to $\mathsf{MaxIP}(x, Y)$. There is also another LSH data structure [65] with longer preprocessing time and larger space that could solve the $(c, \tau)$-MaxIP with similar query time complexity. We refer readers to Section 8.2 in [74] for more details. Moreover, Corrolary B.1 could be applied to projected MaxIP problem.

---

[i]It is obvious to boost probability from constant to $\delta$ by repeating the data structure $\log(1/\delta)$ times.

**Corollary B.2.** *Let $c \in (0,1)$ and $\tau \in (0,1)$. Let $\phi, \psi : \mathbb{R}^d \to \mathbb{R}^k$ denote two transforms. Let $\mathcal{T}_\phi$ denote the time to compute $\phi(x)$ and $\mathcal{T}_\psi$ denote the time to compute $\psi(y)$. Given a set of $n$-points $Y \in \mathbb{R}^d$ with $\psi(Y) \subset \mathcal{S}^{k-1}$ on the sphere, one can construct a data structure with $O(dn^{1+o(1)} + \mathcal{T}_\psi n)$ preprocessing time and $O(n^{1+o(1)} + dn)$ space so that for any query $x \in \mathbb{R}^d$ with $\phi(x) \in \mathcal{S}^{k-1}$, we take query time complexity $O(d \cdot n^\rho + \mathcal{T}_\phi)$ to solve $(c, \phi, \psi, \tau)$-MaxIP with respect to $(x, Y)$ with probability at least $0.9$, where $\rho := \frac{2(1-\tau)^2}{(1-c\tau)^2} - \frac{(1-\tau)^4}{(1-c\tau)^4} + o(1)$.*

*Proof.* The preprocessing phase can be decomposed in two parts.

- It takes $O(\mathcal{T}_\psi n)$ time to transform every $y \in Y$ into $\psi(y)$.

- It takes $O(O(dn^{1+o(1)})$ time and $O(dn^{1+o(1)} + dn)$ to index every $\psi(y)$ into LSH using Corrolary B.1.

The query phase can be decomposed in two parts.

- It takes $O(\mathcal{T}_\phi)$ time to transform every $x \in \mathbb{R}^d$ into $\phi(x)$.

- It takes $O(d \cdot n^\rho)$ time perform query for $\phi(x)$ in LSH using Corrolary B.1.

$\square$

## C  Algorithms

### C.1  Problem Formulation

In this section, we show how to use Frank-Wolfe Algorithm to solve the Problem C.1.

**Problem C.1.**

$$\min_{w \in \mathcal{B}} g(w) \tag{10}$$

*We have the following assumptions:*

- $g : \mathbb{R}^d \to \mathbb{R}$ *is a differentiable function.*

- $S \subset \mathbb{R}^d$ *is a finite feasible set.* $|S| = n$.

- $\mathcal{B} = \mathcal{B}(S) \subset \mathbb{R}^d$ *is the convex hull of the finite set $S \subset \mathbb{R}^d$ defined in Definition A.8.*

- $D_{\max}$ *is the maximum diameter of $\mathcal{B}(S)$ defined in Definition A.8.*

In Problem C.1, function $g$ could have different proprieties about convexity and smoothness.

To solve this problem, we introduce a Frank-Wolfe Algorithm shown in Algorithm 1.

---

**Algorithm 1** Frank-Wolf algorithm for Problem C.1

---

1: **procedure** FRANKWOLFE($S \subset \mathbb{R}^d$)
2:     $T \leftarrow O(\frac{\beta D_{\max}^2}{\epsilon}), \forall t \in [T]$
3:     $\eta \leftarrow \frac{2}{t+2}$
4:     Start with $w^0 \in \mathcal{B}$.                   $\triangleright \mathcal{B} = \mathcal{B}(S)$(see Definition A.8).
5:     **for** $t = 1 \to T - 1$ **do**
6:         $s^t \leftarrow \arg\min_{s \in S} \langle \nabla g(w^t), s \rangle$
7:         $w^{t+1} \leftarrow (1 - \eta_t)w^t + \eta_t s^t$
8:     **end for**
9:     **return** $w^T$
10: **end procedure**

---

One of the major computational bottleneck of Algorithm 1 is the cost paid in each iteration. Algorithm 1 has to linear scan all the $s \in S$ in each iteration. To tackle this issue, we propose a Frank-Wolfe Algorithm with sublinear cost in each iteration.

## C.2  Sublinear Frank-Wolfe Algorithm

In this section, we present the Frank-Wolfe algorithm with sublinear cost per iteration using LSH. The first step is to formulate the line 6 in Algorithm 1 as a projected MaxIP problem defined in Definition A.5. To achieve this, we present a general MaxIP transform.

**Proposition C.2** (MaxIP transform). *Let $\phi_1, \psi_1 : \mathbb{R}^d \to \mathbb{R}^{k_1}$ and $\phi_2, \psi_2 : \mathbb{R}^d \to \mathbb{R}^{k_2}$ to be the projection functions. Given the polynomial function $p(z) = \sum_{i=0}^{D} a_i z^i$, we show that*

$$\langle \phi_1(x), \psi_1(y) \rangle + p(\|\phi_2(x) - \psi_2(y)\|_2^2) = \langle \phi(x), \psi(y) \rangle \tag{11}$$

*where $\phi, \psi : \mathbb{R}^d \to \mathbb{R}^{k_1 + k_2(D+1)^2}$ is the decomposition function.*

*Proof.* Because $\phi_2(x), \psi_2(y) \in \mathbb{R}^{k_2}$, $\|\phi_2(x) - \psi_2(y)\|_2^{2i} = \sum_{j=1}^{k_2} (\phi_2(x)_j - \psi_2(y)_j)^{2i}$. This is the sum over dimensions. Then, we have

$$p(\|\phi_2(x) - \psi_2(y)\|_2^2) = \sum_{i=0}^{D} a_i \|\phi_2(x) - \psi_2(y)\|_2^{2i}$$

$$= \sum_{i=0}^{D} a_i \sum_{j=1}^{k_2} (\phi_2(x)_j - \psi_2(y)_j)^{2i}$$

where the first follows from definition of polynomial $p$, and the second step follows from definition of $\ell_2$ norm.

Here $\phi_2(x)_j$ means the $j$th entry of $\phi_2(x)$. Using the binomial theorem, we decompose $(\phi_2(x)_j - \psi_2(y)_j)^{2i}$ as:

$$(\phi_2(x)_j - \psi_2(y)_j)^{2i}$$

$$= \sum_{l=0}^{2i} \binom{2i}{l} \phi_2(x)_j^{2i-l} \psi_2(y)_j^l$$

$$= \langle \underbrace{[\phi_2(x)_j^{2i}, \cdots, \phi_2(x)_j^{2i-l}, \cdots, \phi_2(x)_j, 1]}_{u_j}, \underbrace{[1, \psi_2(y)_j, \cdots, \psi_2(y)_j^l, \cdots, \psi_2(y)_j^{2i}]}_{v_j} \rangle$$

Then, we generate two vectors $u^i \in \mathbb{R}^{k_2(2i+1)}$ and $v^i \in \mathbb{R}^{k_2(2i+1)}$

$$u^i = \begin{bmatrix} u_1 & \cdots & u_j & \cdots & u_{k_2} \end{bmatrix} \qquad u_j = \begin{bmatrix} \phi_2(x)_j^{2i} & \cdots & \phi_2(x)_j^{2i-l} & \cdots & \phi_2(x)_j & 1 \end{bmatrix}^\top$$

$$v^i = \begin{bmatrix} v_1 & \cdots & v_j & \cdots & v_{k_2} \end{bmatrix} \qquad v_j = \begin{bmatrix} 1 & \psi_2(y)_j & \cdots & \psi_2(y)_j^l & \cdots & \psi_2(y)_j^{2i} \end{bmatrix}^\top$$

Thus, $\sum_{j=1}^{k_2} (\phi_2(x)_j - \psi_2(y)_j)^{2i}$ can be rewrite with inner product by concatenating all the $u_j$ together and then concatenating all the $v_j$.

$$\sum_{j=1}^{k_2} (\phi_2(x)_j - \psi_2(y)_j)^{2i} = \langle u^i, v^i \rangle.$$

We make vectors $b \in \mathbb{R}^{k_2(D+1)^2}$ and $c \in \mathbb{R}^{k_2(D+1)^2}$ such as

$$b = [u^0 \cdots, u^i, \cdots, u_D]$$

$$c = [a_0 v^0, \cdots, a_i v^i, \cdots, a_D v^D]$$

So that

$$\sum_{i=0}^{D} a_i \sum_{j=1}^{k_2} (\phi_2(x)_j - \psi_2(y)_j)^{2i} = \sum_{i=0}^{D} a_i \langle u^i, v^i \rangle = \sum_{i=0}^{D} \langle u^i, a_i v^i \rangle = \langle b, c \rangle$$

Finally, we have

$$\begin{aligned}
\langle \phi_1(x), \psi_1(y) \rangle + p(\|\phi_2(x) - \psi_2(y)\|_2^2) &= \langle \phi_1(x), \psi_1(y) \rangle + \langle b, c \rangle \\
&= \langle [\phi_1(x), b], [\psi_1(y), c] \rangle \\
&= \langle \phi(x), \psi(y) \rangle
\end{aligned}$$

Total projected dimension:

$$\begin{aligned}
k_1 + \sum_{i=0}^{D} k_2(2i+1) &= k_1 + (D+1)k_2 + 2k_2 \sum_{i=1}^{D} i \\
&= k_1 + (D+1)k_2 + 2k_2 \cdot \frac{D(D+1)}{2} \\
&= k_1 + k_2(D+1)^2
\end{aligned}$$

$\square$

Therefore, any binary function with format $\langle \phi_1(x), \psi_1(y) \rangle + p(\|\phi_2(x) - \psi_2(y)\|_2^2)$ defined in Proposition C.2 can be transformed as a inner product.

Next, we show that a modified version of line 6 in Algorithm 1 can be formulated as a projected MaxIP problem.

**Corollary C.3** (Equivalence between projected MaxIP and Min-IP). *Let $g$ be a differential function defined on convex set $\mathcal{K} \subset \mathbb{R}^d$. Given $\eta \in (0,1)$ and $x, y \in \mathcal{K}$, we define $\phi, \psi : \mathbb{R}^d \to \mathbb{R}^{d+3}$ as follows:*

$$\phi(x) := \left[ \frac{\phi_0(x)^\top}{D_x} \quad 0 \quad \sqrt{1 - \frac{\|\phi_0(x)\|_2^2}{D_x^2}} \right]^\top \quad \psi(y) := \left[ \frac{\psi_0(y)^\top}{D_y} \quad \sqrt{1 - \frac{\|\psi_0(y)\|_2^2}{D_y^2}} \quad 0 \right]^\top$$

*where*

$$\phi_0(x) := [\nabla g(x)^\top, x^\top \nabla g(x)]^\top \quad \psi_0(y) := [-y^\top, 1]^\top$$

*, $D_x$ is the maximum diameter of $\phi_0(x)$ and $D_y$ is the maximum diameter of $\psi_0(y)$.*

*Then, for all $x, y \in \mathbb{R}^d$, we transform them into unit vector $\phi(x)$ and $\psi(y)$ on $\mathcal{S}^{d+2}$. Moreover, we have*

$$\langle y - x, \nabla g(x) \rangle = -D_x D_y \langle \phi(x), \psi(y) \rangle$$

*Further, the $(\phi, \psi)$-MaxIP (Definition A.5) is equivalent to the $(\nabla g, 0)$-Min-IP (Definition A.7).*

$$\arg\max_{y \in \mathcal{K}} \langle \phi(x), \psi(y) \rangle = \arg\min_{y \in \mathcal{K}} \langle y - x, \nabla g(x) \rangle$$

*In addition, let $\mathcal{T}_\psi$ denote the time of evaluating at any point $y \in \mathbb{R}^d$ for function $\psi$, then we have $\mathcal{T}_\psi = O(1)$.*

*Let $\mathcal{T}_\phi$ denote the time of evaluating at any point $x \in \mathbb{R}^d$ for function $\phi$, then we have $\mathcal{T}_\phi = \mathcal{T}_{\nabla g} + O(d)$, where the $\mathcal{T}_{\nabla g}$ denote the time of evaluating function $\nabla g$ at any point $x \in \mathbb{R}^d$.*

*Proof.* We start with showing that $\|\phi(x)\|_2 = \|\psi(y)\|_2 = 1$. Next, we show that

$$\begin{aligned}
\langle \phi(x), \psi(y) \rangle &= \frac{\langle \phi_0(x), \psi_0(y) \rangle}{D_x D_y} \\
&= \frac{\langle -y, \nabla g(x) \rangle + \langle x, \nabla g(x) \rangle}{D_x D_y}
\end{aligned}$$

$$= -\frac{\langle y - x, \nabla g(x)\rangle}{D_x D_y}$$

where the first step follows from definition of $\phi$ and $\psi$, the second step follows from definition of $\phi_0$ and $\psi_0$, the last step is a reorganization.

Based on the results above,

$$\arg\max_{y\in\mathcal{K}}\langle\phi(x),\psi(y)\rangle = \arg\min_{y\in\mathcal{K}}\langle y - x, \nabla g(x)\rangle$$

$\square$

Using Corollary C.3, the direction search in Frank-Wolfe algorithm iteration is equivalent to a $(\phi, \psi)$-MaxIP problem. In this way, we propose Algorithm 2, an Frank-Wolfe algorithm with sub-linear cost per iteration using LSH.

---

**Algorithm 2** Sublinear Frank-Wolfe for Problem C.1

---

1: **data structure** LSH                                                          ▷ Corollary B.2
2:     INIT($S \subset \mathbb{R}^d$, $n \in \mathbb{N}$, $d \in \mathbb{N}$, $c \in (0, 1)$)
3:                          ▷ $|S| = n$, $c \in (0, 1)$ is LSH parameter, and $d$ is the dimension of data
4:     QUERY($x \in \mathbb{R}^d$, $\tau \in (0, 1)$)                          ▷ $\tau \in (0, 1)$ is LSH parameter
5: **end data structure**
6:
7: **procedure** SUBLINEARFRANKWOLFE($S \subset \mathbb{R}^d$, $n \in \mathbb{N}$, $d \in \mathbb{N}$, $c \in (0, 1)$, $\tau \in (0, 1)$)          ▷ Theorem D.1
8:     Construct $\phi, \psi : \mathbb{R}^d \to \mathbb{R}^{d+1}$ as Corollary C.3
9:     **static** LSH LSH
10:    LSH.INIT($\psi(S), n, d + 3, c$)
11:    Start with $w^0 \in \mathcal{B}$.                                 ▷ $\mathcal{B} = \mathcal{B}(S)$(see Definition A.8).
12:    $T \leftarrow O(\frac{\beta D_{\max}^2}{c^2 \epsilon})$
13:    $\eta \leftarrow \frac{2}{c(t+2)}, \forall t \in [T]$
14:    **for** $t = 1 \to T - 1$ **do**
15:        /* Query with $w^t$ and retrieve its $(c, \phi, \psi, \tau)$-MaxIP $s^t \in S$ from LSH data structure */
16:        $s^t \leftarrow$ LSH.QUERY($\phi(w^t), \tau$)
17:        /* Update $w^t$ in the chosen direction*/
18:        $w^{t+1} \leftarrow (1 - \eta_t) \cdot w^t + \eta_t \cdot s^t$
19:    **end for**
20:    **return** $w^T$
21: **end procedure**

---

# D   Convergence Analysis

In this Section D, analyze the convergence of our Sublinear Frank-Wolfe algorithm in Algorithm 2 when $g$ is convex (see Definition A.10) and $\beta$-smooth (see Definition A.9). Moreover, we compare our sublinear Frank-Wolfe algorithm with Frank-Wolfe algorithm in Algorithm 1 in terms of number of iterations and cost per iteration.

## D.1   Summary

We first show the comparsion results in Table 2. We list the statement, preprocessing time, number of iterations and cost per iteration for our algorithm and original Frank-Wolfe algorithm to converge. As shown in the table, with $O(dn^{1+o(1)} \cdot \kappa)$ preprocessing time, Algorithm 2 achieves $O(dn^\rho \cdot \kappa + \mathcal{T}_g)$ cost per iteration with $\frac{1}{c^2}$ more iterations.

## D.2   Convergence of Sublinear Frank-Wolfe Algorithm

The goal of this section is to prove Theorem D.1.

| Algorithm | Statement | Preprocessing | #iters | cost per iter |
|---|---|---|---|---|
| Algorithm 1 | [9] | 0 | $O(\beta D_{\max}^2/\epsilon)$ | $O(dn + \mathcal{T}_g)$ |
| Algorithm 2 | Theorem D.1 | $O(dn^{1+o(1)} \cdot \kappa)$ | $O(c^{-2}\beta D_{\max}^2/\epsilon)$ | $O(dn^\rho \cdot \kappa + \mathcal{T}_g)$ |

Table 2: Comparison between original Frank-Wolfe algorithm and our sublinear Frank-Wolfe algorithm. Here $\mathcal{T}_g$ denotes the time for computing gradient of $g$, $c \in (0, 1)$ is the approximation factor of LSH. We let $\kappa := \Theta(\log(T/\delta))$ where $T$ is the number of iterations and $\delta$ is the failure probability. $\rho \in (0, 1)$ is a fixed parameter determined by LSH.

**Theorem D.1** (Convergence result of Sublinear Frank-Wolfe, a formal version of Theorem 3.1).
*Let $g : \mathbb{R}^d \to \mathbb{R}$ denote a convex (see Definition A.10) and $\beta$-smooth function (see Definition A.9). Let the complexity of calculating $\nabla g(x)$ to be $\mathcal{T}_g$. Let $\phi, \psi : \mathbb{R}^d \to \mathbb{R}^k$ denote two transforms in Corollary C.3. Let $S \subset \mathbb{R}^d$ denote a set of points with $|S| = n$, and $\mathcal{B} \subset \mathbb{R}^d$ is the convex hull of $S$ (see Definition A.8). For any parameters $\epsilon, \delta$, there is an iterative algorithm with that takes $O(dn^{1+o(1)} \cdot \kappa)$ preprocessing time and $O((n^{1+o(1)} + dn) \cdot \kappa)$ space, takes $T = O(\frac{\beta D_{\max}^2}{\epsilon})$ iterations and $O(dn^\rho \cdot \kappa + \mathcal{T}_g)$ cost per iteration, starts from a random $w^0$ from $\mathcal{B}$ as initialization point, updates the $w$ in each iteration as follows:*

$$s^t \leftarrow (c, \phi, \psi, \tau)\text{-MaxIP } of \ w^t \ with \ respect \ to \ S$$
$$w^{t+1} \leftarrow w^t + \eta \cdot (s^t - w^t)$$

*and outputs $w^T \in \mathbb{R}^d$ from $\mathcal{B}$ such that*

$$g(w^T) - \min_{w \in \mathcal{B}} g(w) \le \epsilon,$$

*holds with probability at least $1 - \delta$. Here $\kappa := \Theta(\log(T/\delta))$ and $\rho := \frac{2(1-\tau)^2}{(1-c\tau)^2} - \frac{(1-\tau)^4}{(1-c\tau)^4} + o(1)$.*

*Proof.* **Convergence.**

Let $t$ denote some fixed iteration. We consider two cases:

- **Case 1.** $\tau > \max_{s \in S} \langle \psi(s), \phi(w^t) \rangle$;
- **Case 2.** $\tau \le \max_{s \in S} \langle \psi(s), \phi(w^t) \rangle$.

**Case 1.** In this case, we can show that

$$\tau \ge \max_{s \in S} \langle \psi(s), \phi(w^t) \rangle$$
$$\ge \frac{\langle \psi(w^*), \phi(w^t) \rangle}{D_x D_y}$$
$$= \frac{\langle w^t - w^*, \nabla g(w^t) \rangle}{D_x D_y}$$
$$\ge \frac{g(w^t) - g(w^*)}{D_x D_y},$$

where the first step follows from Corollary C.3, the second step follows from the Corollary A.11, the third step is a reorganization, the last step follows the convexity of $g$ (see Definition A.10).

Thus, as long as $\tau \ge D_x D_y \epsilon$, then we have

$$g(w^t) - g(w^*) \le \epsilon.$$

This means we already converges to the $\epsilon$-optimal solution.

**Case 2.** We start with the upper bounding $\langle s^t - w^t, \nabla g(w^t) \rangle$ as

$$\langle s^t - w^t, \nabla g(w^t) \rangle = -D_x D_y \langle \psi(s^t), \phi(w^t) \rangle$$

$$\leq -c \cdot D_x D_y \max_{s \in S} \langle \psi(s), \phi(w^t) \rangle$$
$$\leq -c \cdot D_x D_y \langle \psi(w^*), \phi(w^t) \rangle$$
$$= c \langle w^* - w^t, \nabla g(w^t) \rangle \tag{12}$$

where the first step follows from Corollary C.3, the second step follows from Corollary B.2 and MaxIP condition in Lemma A.12, the third step follows from Corollary A.11.

For convenient of the proof, for each $t$, we define $h_t$ as follows:

$$h_t = g(w^t) - g(w^*). \tag{13}$$

Next, we upper bound $h_{t+1}$ as

$$
\begin{aligned}
h_{t+1} &= g(w^{t+1}) - g(w^*) \\
&= g((1 - \eta_t)w^t + \eta_t s^t) - g(w^*) \\
&\leq g(w^t) + \eta_t \langle s^t - w^t, \nabla g(w^t) \rangle + \frac{\beta}{2} \eta_t^2 \|s^t - w^t\|_2^2 - g(w^*) \\
&\leq g(w^t) + \eta_t \langle s^t - w^t, \nabla g(w^t) \rangle + \frac{\beta D_{\max}^2}{2} \eta_t^2 - g(w^*) \\
&\leq g(w^t) + c\eta_t \langle w^* - w^t, \nabla g(w^t) \rangle + \frac{\beta D_{\max}^2}{2} \eta_t^2 - g(w^*) \\
&= (1 - \eta_t)g(w^t) + c\eta_t \left( g(w^t) + \langle w^* - w^t, \nabla g(w^t) \rangle \right) + \frac{\beta D_{\max}^2}{2} \eta_t^2 - g(w^*) \\
&\leq (1 - \eta_t)g(w^t) + c\eta_t g(w^*) + \frac{\beta D_{\max}^2}{2} \eta_t^2 - g(w^*) \\
&\leq (1 - c\eta_t)g(w^t) - (1 - c\eta_t)g(w^*) + \frac{\beta D_{\max}^2}{2} \eta_t^2 \\
&\leq (1 - c\eta_t)h_t + \frac{\beta D_{\max}^2}{2} \eta_t^2
\end{aligned}
\tag{14}
$$

where the first step follows from definition of $h_{t+1}$ (see Eq. (13)), the second step follows from the update rule of Frank-Wolfe, the third step follows from the definition of $\beta$-smoothness in Definition A.9, the forth step follows from the definition of maximum diameter in Definition A.8, the fifth step follows the Eq (12), the sixth step is a reorganization, the seventh step follows from the definition of convexity (see Definition A.10), the eighth step follows from merging the coefficient of $g(w^*)$, and the last step follows from definition of $h_t$ (see Eq. (13)).

Let $e_t = A_t h_t$, $A_t$ is a parameter and we will decide it later. we have:

$$
\begin{aligned}
e_{t+1} - e_t &= A_{t+1} \left( (1 - c\eta_t)h_t + \frac{\beta D_{\max}^2}{2} \eta_t^2 \right) - A_t h_t \\
&= (A_{t+1}(1 - c\eta_t) - A_t) h_t + \sigma + \frac{\beta D_{\max}^2}{2} A_{t+1} \eta_t^2
\end{aligned}
\tag{15}
$$

Let $A_t = \frac{t(t+1)}{2}$, $c\eta_t = \frac{2}{t+2}$. In this way we rewrite $A_{t+1}(1 - \eta_t) - A_t$ and $A_{t+1} \frac{\eta_t^2}{2}$ as

- $A_{t+1}(1 - \eta_t) - A_t = 0$

- $A_{t+1} \frac{\eta_t^2}{2} = \frac{t+1}{(t+2)c^2} < c^{-2}$

Next, we upper bound $e_{t+1} - e_t$ as:

$$
\begin{aligned}
e_{t+1} - e_t &< 0 + c^{-2} \frac{t+1}{t+2} \beta D_{\max}^2 \\
&< c^{-2} \beta D_{\max}^2
\end{aligned}
\tag{16}
$$

where the first step follows from $A_{t+1}(1 - \eta_t) - A_t = 0$ and $A_{t+1}\frac{\eta_t^2}{2} = \frac{t+1}{(t+2)c^2}$. The second step follows from $\frac{t+1}{t+2} < 1$

Based on Eq (16), we upper bound $e_t$ using induction and have

$$e_t < c^{-2}t\beta D_{\max}^2 \tag{17}$$

Using the definition of $e_t$, we have

$$h_t = \frac{e_t}{A_t} < \frac{2\beta D_{\max}^2}{c^2(t+1)} \tag{18}$$

To make $h_t \leq \epsilon$, $t$ should be in $O(\frac{\beta D_{\max}^2}{c^2 \epsilon})$. Thus, we complete the proof.

**Preprocessing time** According to Corrollary B.2, can construct $\kappa = \Theta(\log(T/\delta))$ LSH data structures for $(c, \phi, \psi, \tau)$-MaxIP with $\phi, \psi$ defined in Corollary C.3. As transforming every $s \in S$ into $\psi(s)$ takes $O(dn)$. Therefore, the total the preprocessing time complexity is $O(dn^{1+o(1)} \cdot \kappa)$.

**Cost per iteration** Given each $w^t$, compute $\nabla g(w^t)$ takes $\mathcal{T}_g$. Next, it takes $O(d)$ time to generate $\phi(w^t)$ according to Corollary C.3 based on $g(w^t)$ and $\nabla g(w^t)$. Next, according to Corollary B.2, it takes $O(dn^\rho \cdot \kappa)$ to retrieve $s^t$ from $\kappa = \Theta(\log(T/\delta))$ LSH data structures. After we select $s^t$, it takes $O(d)$ time to update the $w^{t+1}$. Combining the time for gradient calculation, LSH query and $w^t$ update, the total complexity is $O(dn^\rho \cdot \kappa + \mathcal{T}_g)$ with $\rho := \frac{2(1-\tau)^2}{(1-c\tau)^2} - \frac{(1-\tau)^4}{(1-c\tau)^4} + o(1)$.

$\square$

# E  Herding Algorithm

## E.1  Problem Formulation

In this section, we focus on the Herding algorithm a specific example of Problem C.1. We consider a finite set $\mathcal{X} \subset \mathbb{R}^d$ and a mapping $\Phi : \mathbb{R}^d \to \mathbb{R}^k$. Given a distribution $p(x)$ over $\mathcal{X}$, we denote $\mu \in \mathbb{R}^k$ as

$$\mu = \mathbb{E}_{x \sim p(x)}[\Phi(x)] \tag{19}$$

The goal of Herding algorithm [58] is to find $T$ elements $\{x_1, x_2, \cdots, x_T\} \subseteq \mathcal{X}$ such that $\|\mu - \sum_{t=1}^{T} v_t \Phi(x_t)\|_2$ is minimized. Where $v_t$ is a non-negative weight. The algorithm generates samples by the following:

$$x_{t+1} = \arg\max_{x \in \mathcal{X}} \langle w_t, \Phi(x) \rangle$$
$$w_{t+1} = w_t + \mu - \Phi(x_{t+1}) \tag{20}$$

Let $\mathcal{B}$ denote the convex hull of $X$. [1] show that the recursive algorithm in Eq (20) is equivalent to a Frank-Wolfe algorithm Problem E.1.

**Problem E.1** (Herding)**.**

$$\min_{w \in \mathcal{B}} \frac{1}{2} \|w - \mu\|_2^2$$

*We have the following assumptions:*

- *$S = \Phi(\mathcal{X}) \subset \mathbb{R}^d$ is a finite feasible set. $|S| = n$.*

- *$\mathcal{B} = \mathcal{B}(S) \subset \mathbb{R}^d$ is the convex hull of the finite set $S \subset \mathbb{R}^d$ defined in Definition A.8.*

- *$D_{\max}$ is the maximum diameter of $\mathcal{B}(S)$ defined in Definition A.8*

Therefore, a frank-Wolfe algorithm [1] for Herding is proposed as

---

**Algorithm 3** Herding Algorithm

---

1: **procedure** HERDING($S \subset \mathbb{R}^k$)
2:     $T \leftarrow O(\frac{D_{\max}^2}{\epsilon}), \forall t \in [T]$
3:     $\eta \leftarrow \frac{2}{t+2}$
4:     Start with $w^0 \in \mathcal{B}$.
5:     **for** $t = 1 \to T - 1$ **do**
6:         $s^t \leftarrow \arg\max_{s \in S} \langle w^t - \mu, s \rangle$
7:         $w^{t+1} \leftarrow (1 - \eta)w^t + \eta s^t$
8:     **end for**
9:     **return** $w^T$
10: **end procedure**

---

Algorithm 3 takes $O(nd)$ cost per iteration.

To improve the efficiency of Algorithm 3, we propose a Herding algorithm with sublinear cost per iteration using LSH.

---

**Algorithm 4** Sublinear Herding Algorithm

---

1: **data structure** LSH                                              ▷ Corollary B.2
2:     INIT($S \subset \mathbb{R}^d$, $n \in \mathbb{N}$, $d \in \mathbb{N}$, $c \in (0,1)$)
3:                                      ▷ $|S| = n$, $c \in (0,1)$ is LSH parameter, and $d$ is the dimension of data
4:     QUERY($x \in \mathbb{R}^d$, $\tau \in (0,1)$)                    ▷ $\tau \in (0,1)$ is LSH parameter
5: **end data structure**
6:
7: **procedure** SUBLINEARHERDING($S \subset \mathbb{R}^d$, $n \in \mathbb{N}$, $d \in \mathbb{N}$, $c \in (0,1)$, $\tau \in (0,1)$ )
8:                                                                       ▷ Theorem E.3
9:     Construct $\phi, \psi : \mathbb{R}^d \to \mathbb{R}^{d+1}$ as Corollary C.3
10:    **static** LSH LSH
11:    LSH.INIT($\psi(S), n, d+3, c$)
12:    Start with $w^0 \in \mathcal{B}$.                               ▷ $\mathcal{B} = \mathcal{B}(S)$(see Definition A.8).
13:    $T \leftarrow O(\frac{\beta D_{\max}^2}{c^2 \epsilon})$, $\forall t \in [T]$
14:    $\eta \leftarrow \frac{2}{c(t+2)}$
15:    **for** $t = 1 \to T - 1$ **do**
16:        /* Query with $w^t$ and retrieve its $(c, \phi, \psi)$-MaxIP $s^t \in S$ from LSH data structure */
17:        $s^t \leftarrow$ LSH.QUERY($\phi(w^t), \tau$)
18:        /* Update $w^t$ in the chosen direction*/
19:        $w^{t+1} \leftarrow (1 - \eta_t) \cdot w^t + \eta_t \cdot s^t$
20:    **end for**
21:    **return** $w^T$
22: **end procedure**

---

## E.2   Convergence Analysis

The goal of this section is to show the convergence analysis of our Algorithm 4 compare it with Algorithm 3 for Herding.

We first show the comparison results in Table 3. In this table, we list the statement, preprocessing time, number of iterations and cost per iteration for our algorithm and original Herding algorithm to converge.

| Algorithm | Statement | Preprocessing | #iters | cost per iter |
|---|---|---|---|---|
| Algorithm 3 | [1] | 0 | $O(D_{\max}^2 / \epsilon)$ | $O(dn)$ |
| Algorithm 4 | Theorem E.3 | $O(dn^{1+o(1)} \cdot \kappa)$ | $O(c^{-2} D_{\max}^2 / \epsilon)$ | $O(dn^\rho \cdot \kappa)$ |

Table 3: Comparison between Algorithm 4 and Algorithm 3

Next, we analyze the smoothness of $\frac{1}{2}\|w - \mu\|_2^2$.

**Lemma E.2.** *We show that $g(w) = \frac{1}{2}\|w^T - \mu\|_2^2$ is a convex and $1$-smooth function.*

*Proof.*

$$
\begin{aligned}
g(x) + \langle \nabla g(x), y - x \rangle + \frac{1}{2}\|y - x\|_2^2 &= \frac{1}{2}\|x - \mu\|_2^2 + \langle x - \mu, y - x \rangle + \frac{1}{2}\|y - x\|_2^2 \\
&= \frac{1}{2}(x^\top x - 2x^\top \mu + \mu^\top \mu) + (x^\top y - y^\top \mu \\
&= \frac{1}{2}y^\top y - y^\top \mu + \frac{1}{2}\mu^\top \mu \\
&= \frac{1}{2}\|y - \mu\|_2^2 \\
&= g(y) \tag{21}
\end{aligned}
$$

where all the steps except the last step are reorganizations. The last step follows $g(y) = \frac{1}{2}\|y - \mu\|_2^2$

Rewrite the Eq (21) above, we have

$$g(y) = g(x) + \langle \nabla g(x), y - x \rangle + \frac{1}{2} \|y - x\|_2^2 \tag{22}$$

$$\geq g(x) + \langle \nabla g(x), y - x \rangle \tag{23}$$

$g(x) = \frac{1}{2} \|x - \mu\|_2^2$ is a convex function.

Rewrite the Eq (21) above again, we have

$$g(y) = g(x) + \langle \nabla g(x), y - x \rangle + \frac{1}{2} \|y - x\|_2^2 \tag{24}$$

$$\leq g(x) + \langle \nabla g(x), y - x \rangle + \frac{1}{2} \|y - x\|_2^2 \tag{25}$$

$g(x) = \frac{1}{2} \|x - \mu\|_2^2$ is a 1-smooth convex function.

$\square$

Next, we show the convergence results of Algorithm 4.

**Theorem E.3** (Convergence result of Sublinear Herding, a formal version of Theorem 3.2). *For any parameters $\epsilon, \delta$, there is an iterative algorithm (Algorithm 4) for Problem E.1 that takes $O(dn^{1+o(1)} \cdot \kappa)$ time in pre-processing and $O((n^{1+o(1)} + dn) \cdot \kappa)$ space, takes $T = O(\frac{D_{\max}^2}{c^2 \epsilon})$ iterations and $O(dn^\rho \cdot \kappa)$ cost per iteration, starts from a random $w^0$ from $\mathcal{B}$ as initialization point, updates the $w$ in each iteration and outputs $w^T \in \mathbb{R}^d$ from $\mathcal{B}$ such that*

$$\frac{1}{2} \|w^T - \mu\|_2^2 - \min_{w \in \mathcal{B}} \frac{1}{2} \|w - \mu\|_2^2 \leq \epsilon,$$

*holds with probability at least $1 - \delta$. Here $\rho := \frac{2(1-\tau)^2}{(1-c\tau)^2} - \frac{(1-\tau)^4}{(1-c\tau)^4} + o(1)$ and $\kappa := \Theta(\log(T/\delta))$.*

*Proof.* First, we show that $g(w) = \frac{1}{2} \|w^T - \mu\|_2^2$ is a convex and 1-smooth function. using Lemma E.2. Then, we could prove the theorem using Theorem E.3. Following the fact that the computation of gradient is $O(d)$, we could also provide the query time, preprocesisng time and space complexities.

$\square$

### E.3  Discussion

We show that our sublinear Frank-Wolfe algorithm demonstrated in Algorithm 4 breaks the linear cost per iteration of current Frank-Wolfe algorithm in Algorithm 3 in the Herding algorithm. Meanwhile, the extra number of iterations Algorithm 4 pay is affordable.

Our results show the connection between the extra number of iterations and the cost reduction at each iteration. It represents a formal combination of LSH data structures and Herding algorithm. We hope that this demonstration would provide insights for more applications of LSH in kernel methods and graphical models.

## F  Policy Gradient Optimization

We present the our results on policy gradient in this section.

### F.1  Problem Formulation

In this paper, we focus on the action-constrained Markov Decision Process (ACMDP). In this setting, we are provided with a state $\mathcal{S} \in \mathbb{R}^k$ and action space $\mathcal{A} \in \mathbb{R}^d$, which is the convex hull of $n$-vector. However, at each step $t \in \mathbb{N}$, we could only access a finite subset of actions $\mathcal{C}(s) \subset \mathcal{A}$ with cardinality $n$. Let us assume the $\mathcal{C}(s)$ remains the same in each step. Let us denote $D_{\max}$ as the maximum diameter of $\mathcal{A}$.

When you play with this ACMDP, the policy you choose is defined as $\pi_\theta(s) : \mathcal{S} \to \mathcal{A}$ with parameter $\theta$. Meanwhile, there exists a reward function $r : \mathcal{S} \times \mathcal{A} \in [0, 1]$. Next, we define the Q function as below,

$$Q(s, a|\pi_\theta) = \mathbb{E}\Big[\sum_{t=0}^\infty \gamma^t r(s_t, a_t)|s_0 = s, a_0 = a, \pi_\theta\Big].$$

where $\gamma \in (0, 1)$ is a discount factor.

Given a state distribution $\mu$, the objective of policy gradient is to maximize the expected value $J(\mu, \pi_\theta) = \mathbb{E}_{s\sim\mu, a\sim\pi_\theta}[Q(s, a|\pi_\theta)]$ via policy gradient [59] denoted as:

$$\nabla_\theta J(\mu, \pi_\theta) = \mathbb{E}_{s\sim d_\mu^\pi}\Big[\nabla_\theta \pi_\theta(s)\nabla_a Q(s, \pi_\theta(s)|\pi_\theta)|\Big].$$

[5] propose an iterative algorithm that perform MaxIP at each iteration $k$ over actions to find

$$g_k(s) = \max_{a\in\mathcal{C}(s)}\langle a_s^k - \pi_\theta^k(s), \nabla_a Q(s, \pi_\theta^k(s)|\pi_\theta^k))\rangle. \tag{26}$$

Moreover, [5] also have the following statement

**Lemma F.1** ([5]). *Given a ACMDP and the gap $g_k(s)$ in Eq.(26), we show that*

$$J(\mu, \pi_\theta^{k+1}) \geq J(\mu, \pi_\theta^k(s)) + \frac{(1-\gamma)^2\mu_{\min}^2}{2LD_{\max}^2}\sum_{s\in\mathcal{S}} g_k(s)^2$$

Therefore, [5] maximize the expected value via minimizing $g_k(s)$.

In this work, we accelerate Eq. (6) using $(c, \phi, \psi, \tau)$-MaxIP. Here define $\phi : \mathcal{S} \times \mathbb{R}^d \to \mathbb{R}^{d+2}$ and $\psi : \mathbb{R}^d \to \mathbb{R}^{d+3}$ as follows:

**Corollary F.2** (Transformation for policy gradient). *Let $g$ be a differential function defined on convex set $\mathcal{K} \subset \mathbb{R}^d$ with maximum diameter $D_\mathcal{K}$. For any $x, y \in \mathcal{K}$, we define $\phi, \psi : \mathbb{R}^d \to \mathbb{R}^{d+3}$ as follows:*

$$\phi(x) := \begin{bmatrix} \frac{\phi_0(x)^\top}{D_x} & 0 & \sqrt{1 - \frac{\|\phi_0(x)\|_2^2}{D_x^2}} \end{bmatrix}^\top \quad \psi(y) := \begin{bmatrix} \frac{\psi_0(y)^\top}{D_y} & \sqrt{1 - \frac{\|\psi_0(y)\|_2^2}{D_y^2}} & 0 \end{bmatrix}^\top$$

*where*

$$\phi_0(s, \pi_\theta^k) := [\nabla_a Q(s, \pi_\theta^k(s)|\pi_\theta^k)^\top, (\pi_\theta^k)^\top Q(s, \pi_\theta^k(s)|\pi_\theta^k)]^\top$$
$$\psi_0(a) = [a^\top, -1]^\top$$

*and $D_x$ is the maximum diameter of $\phi_0(x)$ and $D_y$ is the maximum diameter of $\psi_0(y)$.*

*Then, for all $x, y \in \mathcal{K}$ we have $g_k(s) = D_x D_y\langle\phi(s, \pi_\theta^k), \psi(a)\rangle$. Moreover, $\phi(x)$ and $\psi(y)$ are unit vectors with norm $1$.*

*Proof.* We show that

$$\langle\phi(s, \pi_\theta^k), \psi(a)\rangle = D_x^{-1}D_y^{-1}\langle\nabla_a Q(s, \pi_\theta^k(s)|\pi_\theta^k), a\rangle - \langle\nabla_a Q(s, \pi_\theta^k(s)|\pi_\theta^k), \pi_\theta^k\rangle$$
$$= D_x^{-1}D_y^{-1}\langle a_s^k - \pi_\theta^k(s), \nabla_a Q(s, \pi_\theta^k(s)|\pi_\theta^k))\rangle$$

where the first step follows the definition of $\phi$ and $\psi$, the second step is an reorganization. □

In this way, we propose a sublinear iteration cost algorithm for policy gradient in Algorithm 5.

---

**Algorithm 5** Sublinear Frank-Wolfe Policy Optimization (SFWPO)

---

1: **data structure** LSH                                           ▷ Corollary B.2
2:      INIT($S \subset \mathbb{R}^d$, $n \in \mathbb{N}$, $d \in \mathbb{N}$, $c \in (0,1)$)
3:                                ▷ $|S| = n$, $c \in (0,1)$ is LSH parameter, and $d$ is the dimension of data
4:      QUERY($x \in \mathbb{R}^d$, $\tau \in (0,1)$)                          ▷ $\tau \in (0,1)$ is LSH parameter
5: **end data structure**
6:
7: **procedure** SFWPO($\mathcal{S} \subset \mathbb{R}^k$, $c \in (0,1)$, $\tau \in (0,1)$)
8:                                                 ▷ Theorem F.3
9:      **Input:** Initialize the policy parameters as $\theta_0 \in R^l$ that satisfies $\pi_\theta^0(s) \in \mathcal{C}(s)$ for all $s \in \mathcal{S}$
10:      **for** each State $s \in \mathcal{S}$ **do**
11:          Construct $\phi, \psi : \mathbb{R}^d \to \mathbb{R}^{d+1}$ as Corollary F.2
12:          **static** LSH LSH$_s$
13:          LSH$_s$ INIT($\psi(\mathcal{C}(s)), n, d+3, c$)
14:      **end for**
15:      $T \leftarrow O(\frac{c^{-2} L D_{\max}^2}{\epsilon^2(1-\gamma)^3 \mu_{\min}^2}$
16:      **for** each iteration $k = 0, 1, \cdots, T$ **do**
17:          **for** each State $s \in \mathcal{S}$ **do**
18:              Use policy $\pi_\theta^k$ and obtain $Q(s, \pi_\theta^k(s) | \pi_\theta^k)$
19:          **end for**
20:          **for** each State $s \in \mathcal{S}$ **do**
21:              $\widehat{a_s^k} \leftarrow$ LSH$_s$.QUERY($\phi(s, \pi_\theta^k(s), \tau)$)
22:              $\widehat{g_k}(s) = \langle \widehat{a_s^k} - \pi_\theta^k(s), \nabla_a Q(s, \pi_\theta^k(s) | \pi_\theta^k)) \rangle$
23:              $\alpha_k(s) = \frac{(1-\gamma)\mu_{\min}}{L D_s^2} \widehat{g_k}(s)$
24:              $\pi_\theta^{k+1}(s) = \pi_\theta^k(s) + \alpha_k(s)(\widehat{a_s^k} - \pi_\theta^k(s))$
25:          **end for**
26:      **end for**
27:      **return** $\pi_\theta^T(s)$
28: **end procedure**

---

## F.2 Convergence Analysis

The goal of this section is to show the convergence analysis of of Algorithm 5 compare it with [5].We first show the comparison results in Table 4.

| Algorithm | Statement | Preprocessing | #iters | cost per iter |
|---|---|---|---|---|
| [5] | [5] | 0 | $O(\frac{\beta D_{\max}^2}{\epsilon^2(1-\gamma)^3 \mu_{\min}^2})$ | $O(dn + \mathcal{T}_Q)$ |
| Algorithm 5 | Theorem F.3 | $O(dn^{1+o(1)} \cdot \kappa)$ | $O(\frac{c^{-2}\beta D_{\max}^2}{\epsilon^2(1-\gamma)^3 \mu_{\min}^2})$ | $O(dn^\rho \cdot \kappa + \mathcal{T}_Q)$ |

Table 4: Comparison between our sublinear policy gradient (Algorithm 5) and [5].

The goal of this section is to prove Theorem F.3.

**Theorem F.3** (Sublinear Frank-Wolfe Policy Optimization (SFWPO), a formal version of Theorem 3.3). *Let $\mathcal{T}_Q$ denote the time for computing the policy graident. Let $D_{\max}$ denote the maximum diameter of action space and $\beta$ is a constant. Let $\gamma \in (0,1)$. Let $\rho \in (0,1)$ denote a fixed parameter. Let $\mu_{\min}$ denote the minimal density of states in $\mathcal{S}$. There is an iterative algorithm (Algorithm 5) that spends $O(dn^{1+o(1)} \cdot \kappa)$ time in preprocessing and $O((n^{1+o(1)} + dn) \cdot \kappa)$ space, takes $O(\frac{\beta D_{\max}^2}{\epsilon^2(1-\gamma)^3 \mu_{\min}^2})$ iterations and $O(dn^\rho \cdot \kappa + \mathcal{T}_Q)$ cost per iterations, start from a random point $\pi_\theta^0$ as initial point, and output policy $\pi_\theta^T$ that is have average gap $\sqrt{\sum_{s \in \mathcal{S}} g_T(s)^2} < \epsilon$ holds with probability at least $1 - 1/\text{poly}(n)$, where $g_T(s)$ is defined in Eq. (26) and $\kappa := \Theta(\log(T/\delta))$.*

*Proof.* Let $\widehat{a_s^k}$ denote the action retrieved by LSH. Note that similar to Case 1 of Theorem D.1, the algorithms convergences if parameter $\tau$ is greater than maximum inner product. Therefore, we

could direct focus on Case 2 and lower bound $\widehat{g}_k(s)$ as

$$
\begin{aligned}
\widehat{g}_k(s) &= \langle \widehat{a_s^k} - \pi_\theta^k(s), \nabla_a Q(s, \pi_\theta^k(s)|\pi_\theta^k)) \rangle \\
&= D_x D_y \langle \phi(s, \pi_\theta^k), \psi(\widehat{a_s^k}) \rangle \\
&\geq c D_x D_y \max_{a \in \mathcal{C}(a)} \langle \phi(s, \pi_\theta^k), \psi(a) \rangle \\
&= c\langle a_s^k, \nabla_a Q(s, \pi_\theta^k(s)|\pi_\theta^k)) \rangle - c\langle \pi_\theta^k(s), \nabla_a Q(s, \pi_\theta^k(s)|\pi_\theta^k)) \rangle \\
&= c g_k(s)
\end{aligned}
\tag{27}
$$

where the first step follows from the line 22 in Algorithm 5, the second step follows from Corollary F.2, the third step follows from Corollary B.2, the forth step follows from Corollary F.2 and the last step is a reorganization.

Next, we upper bound $J(\mu, \pi_\theta^{k+1})$ as

$$
\begin{aligned}
J(\mu, \pi_\theta^{k+1}) &\geq J(\mu, \pi_\theta^k(s)) + \frac{(1-\gamma)^2 \mu_{\min}^2}{2 L D_{\max}^2} \sum_{s \in \mathcal{S}} \widehat{g}_k(s)^2 \\
&\geq J(\mu, \pi_\theta^k(s)) + \frac{c^2(1-\gamma)^2 \mu_{\min}^2}{2 L D_{\max}^2} \sum_{s \in \mathcal{S}} g_k(s)^2
\end{aligned}
\tag{28}
$$

where the first step follows from Lemma F.1, the second step follows from Eq. (27)

Using induction from 1 to $T$, we have

$$
J(\mu, \pi_\theta^T) = J(\mu, \pi_\theta^1) + \frac{c^2(1-\gamma)^2 \mu_{\min}^2}{2 L D_{\max}^2} \sum_{k=0}^{T} \sum_{s \in \mathcal{S}} g_k(s)^2
\tag{29}
$$

Let $G = \sum_{k=0}^{T} \sum_{s \in \mathcal{S}} g_k(s)^2$, we upper bound $G$ as

$$
\begin{aligned}
G &\leq \frac{2 L D_{\max}^2}{c^2(1-\gamma)^2 \mu_{\min}^2} (J(\mu, \pi_\theta^T) - J(\mu, \pi_\theta^0)) \\
&\leq \frac{2 L D_{\max}^2}{c^2(1-\gamma)^2 \mu_{\min}^2} J(\mu, \pi_\theta^*)) \\
&\leq \frac{2 L D_{\max}^2}{c^2(1-\gamma)^3 \mu_{\min}^2}
\end{aligned}
\tag{30}
$$

where the first step follows from Eq (29), the second step follows from $J(\mu, \pi_\theta^*) \geq J(\mu, \pi_\theta^T)$, last step follows from $J(\mu, \pi_\theta^*) \leq (1-\gamma)^{-1}$.

Therefore, we upper bound $\sum_{s \in \mathcal{S}} g_T(s)^2$ as

$$
\begin{aligned}
\sum_{s \in \mathcal{S}} g_T(s)^2 &\leq \frac{1}{T+1} G \\
&\leq \frac{1}{T+1} \frac{2 L D_{\max}^2}{c^2(1-\gamma)^3 \mu_{\min}^2}
\end{aligned}
\tag{31}
$$

where the first step is a reorganization, the second step follows that $\sum_{s \in \mathcal{S}} g_T(s)^2$ is non-increasing, the second step follows from Eq (30).

If we want $\sum_{s \in \mathcal{S}} g_T(s)^2 < \epsilon^2$, $T$ should be $O(\frac{c^{-2} L D_{\max}^2}{\epsilon^2 (1-\gamma)^3 \mu_{\min}^2})$

**Preprocessing time** According to Corrollary B.2, can construct $\kappa = \Theta(\log(T/\delta))$ LSH data structures for $(c, \phi, \psi, \tau)$-MaxIP with $\phi, \psi$ defined in Corollary F.2. As transforming every $a \in \mathcal{A}$ into $\psi(a)$ takes $O(dn)$. Therefore, the total the preprocessing time complexity is $O(dn^{1+o(1)} \cdot \kappa)$.

**Cost per iteration** Given each $w^t$, compute the policy gradient takes $\mathcal{T}_Q$. Next, it takes $O(d)$ time to generate $\phi(s, \pi_\theta^k)$ according to Corollary C.3 based on policy gradient. Next, according to Corrollary B.2, it takes $O(dn^\rho \cdot \kappa)$ to retrieve action from $\kappa = \Theta(\log(T/\delta))$ LSH data structures. After we select action, it takes $O(d)$ time to compute the gap the update the value. Thus, the total complexity is $O(dn^\rho \cdot \kappa + \mathcal{T}_Q)$ with $\rho := \frac{2(1-\tau)^2}{(1-c\tau)^2} - \frac{(1-\tau)^4}{(1-c\tau)^4} + o(1)$.

$\square$

### F.3 Discussion

We show that our sublinear Frank-Wolfe based policy gradient algorithm demonstrated in Algorithm 5 breaks the linear cost per iteration of current Frank-Wolfe based policy gradient algorithm algorithm. Meanwhile, the extra number of iterations Algorithm 5 pay is affordable.

Our results extends the LSH to policy gradient optimization and characterize the relationship between the more iterations we paid and the cost we save at each iteration. These results indicates a formal combination of LSH data structures and policy gradient optimization with theoretical gurantee. We hope that this demonstration would provide new research directions for more applications of LSH in robotics and planning.

## G   More Data Structures: Adaptive MaxIP Queries

In optimization, the gradient at each iteration is not independent from the previous gradient. Therefore, it becomes a new setting for using $(c, \tau)$-MaxIP. If we take the gradient as query and the feasible set as the data set, the queries in each step forms an adaptive sequence. In this way, the failure probability of LSH or other $(c, \tau)$-MaxIP data-structures could not be union bounded. To extend $(c, \tau)$-MaxIP data-structures such as LSH and graphs into this new setting, we use a query quantization method. This method is standard in various machine learning tasks [56, 74].

We start with relaxing the $(c, \tau)$-MaxIP with a inner product error.

**Definition G.1** (Relaxed approximate MaxIP). *Let approximate factor $c \in (0, 1)$ and threshold $\tau \in (0, 1)$. Let $\lambda \geq 0$ denote an additive error. Given an $n$-vector set $Y \subset \mathbb{S}^{d-1}$, the objective of $(c, \tau, \lambda)$-MaxIP is to construct a data-structure that, for a query $x \in \mathbb{S}^{d-1}$ with conditions that $\max_{y \in Y} \langle x, y \rangle \geq \tau$, it retrieves vector $z \in Y$ that $\langle x, z \rangle \geq c \cdot \max_{y \in Y} \langle x, y \rangle - \lambda$.*

Then, we present a query quantization approach to solve $(c, \tau, \lambda)$-MaxIP for adaptive queries. We assume that the $Q$ is the convex hull of all queries. For any query $x \in Q$, we perform a quantization on it and locate it to the nearest lattice with center $\widehat{q} \in Q$. Here the lattice has maximum diameter $2\lambda$. Then, we query $\widehat{q}$ on data-structures e.g., LSH, graphs, alias tables. This would generate a $\lambda$ additive error to the inner product. Because the lattice centers are independent, the cumulative failure probability for adaptive query sequence could be union bounded. Formally, we present the corollary as

**Corollary G.2** (A query quantization version of Corollary B.1). *Let approximate factor $c \in (0, 1)$ and threshold $\tau \in (0, 1)$. Given a $n$-vector set $Y \subset \mathbb{S}^{d-1}$, there exits a data-structure with $O(dn^{1+o(1)} \cdot \kappa)$ preprocessing time and $O((n^{1+o(1)} + dn) \cdot \kappa)$ space so that for every query $x$ in an adaptive sequence $X = \{x_1, x_2, \cdots, x_T\} \subset \mathbb{S}^{d-1}$, we take $O(dn^\rho \cdot \kappa)$ query time to solve $(c, \tau, \lambda)$-MaxIP with respect to $(x, Y)$ with probability at least $1 - \delta$, where $\rho = \frac{2(1-\tau)^2}{(1-c\tau)^2} - \frac{(1-\tau)^4}{(1-c\tau)^4} + o(1)$, $\kappa := d \log(ndD_X/(\lambda\delta))$ and $D_X$ is the maximum diameter of all queries in $X$.*

*Proof.* The probability that at least one query $x \in X$ fails is equivalent to the probability that at least one query $\widehat{q} \in \widehat{Q}$ fails. Therefore, we could union bound the probability as:

$$\Pr[\exists \widehat{q} \in \widehat{Q} \ \text{ s.t all } \ (c, \tau)\text{-MaxIP}(q, \widehat{Q}) \text{ fail}] = n \cdot (dD_X/\lambda)^d \cdot (1/10)^\kappa \leq \delta$$

where the second step follows from $\kappa := d \log(ndD_X/(\lambda\delta))$.

The results of $\widehat{q}$ has a $\lambda$ additive error to the original query. Thus, our results is a $(c, \tau, \lambda)$-MaxIP solution. The time and space complexty is obtained via Corollary B.1. Thus we finish the proof. $\square$

**Definition G.3** (Quantized projected approximate MaxIP). *Let approximate factor $c \in (0,1)$ and threshold $\tau \in (0,1)$. Let $\lambda \geq 0$ denote an additive error. Let $\phi, \psi : \mathbb{R}^d \to \mathbb{R}^k$ denote two transforms. Given an $n$-point dataset $Y \subset \mathbb{R}^d$ so that $\psi(Y) \subset \mathbb{S}^{k-1}$, the goal of the $(c, \phi, \psi, \tau, \lambda)$-MaxIP is to build a data structure that, given a query $x \in \mathbb{R}^d$ and $\phi(x) \in \mathbb{S}^{k-1}$ with the promise that $\max_{y \in Y} \langle \phi(x), \psi(y) \rangle \geq \tau - \lambda$, it retrieves a vector $z \in Y$ with $\langle \phi(x), \psi(z) \rangle \geq c \cdot (\phi, \psi)$-MaxIP$(x, Y)$.*

Next, we extend Corollary G.2 to adaptive queries.

**Corollary G.4.** *Let $c \in (0,1)$, $\tau \in (0,1)$, $\lambda \geq 0$ and $\delta \geq 0$. Let $\phi, \psi : \mathbb{R}^d \to \mathbb{R}^k$ denote two transforms. Let $\mathcal{T}_\phi$ denote the time to compute $\phi(x)$ and $\mathcal{T}_\psi$ denote the time to compute $\psi(y)$. Given a set of $n$-points $Y \in \mathbb{R}^d$ with $\psi(Y) \subset \mathcal{S}^{k-1}$ on the sphere, there exists a data structure with $O(dn^{1+o(1)} \cdot \kappa + \mathcal{T}_\psi n)$ preprocessing time and $O((dn^{1+o(1)} + dn) \cdot \kappa)$ space so that for any query $x \in \mathbb{R}^d$ with $\phi(x) \in \mathcal{S}^{k-1}$, we take $O(dn^\rho \cdot \kappa + \mathcal{T}_\phi)$ query time to solve $(c, \phi, \psi, \tau, \lambda)$-MaxIP with respect to $(x, Y)$ with probability at least $1 - \delta$, where $\rho := \frac{2(1-\tau)^2}{(1-c\tau)^2} - \frac{(1-\tau)^4}{(1-c\tau)^4} + o(1)$, $\kappa := d\log(ndD_X/(\lambda\delta))$ and $D_X$ is the maximum diameter of all queries in $X$.*

Finally, we present a modified version of Theorem D.1.

**Theorem G.5** (Convergence result of Frank-Wolfe via LSH with adaptive input). *Let $g : \mathbb{R}^d \to \mathbb{R}$ denote a convex (see Definition A.10) and $\beta$-smooth function (see Definition A.9). Let the complexity of calculating $\nabla g(x)$ to be $\mathcal{T}_g$. Let $S \subset \mathbb{R}^d$ denote a set of points with $|S| = n$, and $\mathcal{B} \subset \mathbb{R}^d$ is the convex hull of $S$ defined in Definition A.8. For any parameters $\epsilon, \delta$, there is an iterative algorithm with $(c, \phi, \psi, \tau, c^{-2}\epsilon/4)$-MaxIP data structure that takes $O(dn^{1+o(1)} \cdot \kappa)$ preprocessing time and $O((n^{1+o(1)} + dn) \cdot \kappa)$ space, takes $T = O(\frac{\beta D_{\max}^2}{\epsilon})$ iterations and $O(dn^\rho \cdot \kappa + \mathcal{T}_g)$ cost per iteration, starts from a random $w^0$ from $\mathcal{B}$ as initialization point, updates the $w$ in each iteration as follows:*

$$s^t \leftarrow (c, \phi, \psi, \tau, c^{-2}\epsilon/4)\text{-MaxIP of } w^t \text{ with respect to } S$$
$$w^{t+1} \leftarrow w^t + \eta \cdot (s^t - w^t)$$

*and outputs $w^T \in \mathbb{R}^d$ from $\mathcal{B}$ such that*

$$g(w^T) - \min_{w \in \mathcal{B}} g(w) \leq \epsilon,$$

*holds with probability at least $1 - \delta$. Here $\kappa := d\log(ndD_X/(\lambda\delta))$ and $\rho := \frac{2(1-\tau)^2}{(1-c\tau)^2} - \frac{(1-\tau)^4}{(1-c\tau)^4} + o(1)$.*

*Proof.* **Convergence.**

We start with modifying Eq. (14) with additive MaxIP error $\lambda$ and get

$$h_{t+1} = (1 - c\eta_t)h_t + \frac{\beta D_{\max}^2}{2}\eta_t^2 + \eta_t \lambda$$

Let $e_t = A_t h_t$ with $A_t = \frac{t(t+1)}{2}$. Let $\eta_t = \frac{2}{c(t+2)}$. Let $\lambda = \frac{\beta D_{\max}^2}{T+1}$ Following the proof in Theorem D.1, we upper bound $e_{t+1} - e_t$ as

$$e_{t+1} - e_t \leq \left(A_{t+1}(1 - c\eta_t) - A_t\right)h_t + \frac{\beta D_{\max}^2}{2}A_{t+1}\eta_t^2 + A_{t+1}\eta_t\lambda \tag{32}$$

where

- $A_{t+1}(1 - \eta_t) - A_t = 0$

- $A_{t+1}\frac{\eta_t^2}{2} = \frac{t+1}{(t+2)c^2} < c^{-2}$

- $A_{t+1}\eta_t\lambda = (t+1)\lambda < \beta D_{\max}^2$.

Therefore,

$$e_{t+1} - e_t < 2c^{-2}\beta D_{\max}^2 \tag{33}$$

Based on Eq (33), we upper bound $e_t$ using induction and have

$$e_t < 2c^{-2}t\beta D_{\max}^2 \tag{34}$$

Using the definition of $e_t$, we have

$$h_t = \frac{e_t}{A_t} < \frac{4\beta D_{\max}^2}{c^2(t+1)} \tag{35}$$

To make $h_T \leq \epsilon$, $T$ should be in $O(\frac{\beta D_{\max}^2}{c^2\epsilon})$. Moreover, $\lambda = \frac{\beta D_{\max}^2}{T+1} = \frac{\epsilon}{4c^2}$.

**Preprocessing time** According to Corollary G.4, can construct $\kappa = d\log(ndD_X/(\lambda\delta))$ LSH data structures for $(c, \phi, \psi, \tau, c^{-2}\epsilon/4)$-MaxIP with $\phi, \psi$ defined in Corollary C.3. As transforming every $s \in S$ into $\psi(s)$ takes $O(dn)$. Therefore, the total the preprocessing time complexity is $O(dn^{1+o(1)} \cdot \kappa)$ and space complexity is $O((n^{1+o(1)} + dn) \cdot \kappa)$.

**Cost per iteration** Given each $w^t$, compute $\nabla g(w^t)$ takes $\mathcal{T}_g$. Next, it takes $O(d)$ time to generate $\phi(w^t)$ according to Corollary C.3 based on $g(w^t)$ and $\nabla g(w^t)$. Next, according to Corollary G.4, it takes $O(dn^\rho \cdot \kappa)$ to retrieve $s^t$ from $\kappa$ LSH data structures. After we select $s^t$, it takes $O(d)$ time to update the $w^{t+1}$. Combining the time for gradient calculation, LSH query and $w^t$ update, the total complexity is $O(dn^\rho \cdot \kappa + \mathcal{T}_g)$ with $\rho := \frac{2(1-\tau)^2}{(1-c\tau)^2} - \frac{(1-\tau)^4}{(1-c\tau)^4} + o(1)$.

$\square$

Similarly, we could extend the results to statements of Herding algorithm and policy gradient.

**Theorem G.6** (Modified result of Sublinear Herding). *For any parameters $\epsilon, \delta$, there is an iterative algorithm (Algorithm 4) for Problem E.1 with $c^{-2}\epsilon/4$ query quantization that takes $O(dn^{1+o(1)} \cdot \kappa)$ time in pre-processing and $O((n^{1+o(1)}+dn)\cdot\kappa)$ space, takes $T = O(\frac{D_{\max}^2}{c^2\epsilon})$ iterations and $O(dn^\rho \cdot \kappa)$ cost per iteration, starts from a random $w^0$ from $\mathcal{B}$ as initialization point, updates the $w$ in each iteration based on Algorithm 4 and outputs $w^T \in \mathbb{R}^d$ from $\mathcal{B}$ such that*

$$\frac{1}{2}\|w^T - \mu\|_2^2 - \min_{w \in \mathcal{B}} \frac{1}{2}\|w - \mu\|_2^2 \leq \epsilon,$$

*holds with probability at least $1 - \delta$. Here $\rho := \frac{2(1-\tau)^2}{(1-c\tau)^2} - \frac{(1-\tau)^4}{(1-c\tau)^4} + o(1)$ and $\kappa := d\log(ndD_X/(\lambda\delta))$.*

**Theorem G.7** (Modified result of Sublinear Frank-Wolfe Policy Optimization (SFWPO)). *Let $\mathcal{T}_Q$ denote the time for computing the policy graident. Let $D_{\max}$ denote the maximum diameter of action space and $\beta$ is a constant. Let $\gamma \in (0,1)$. Let $\rho \in (0,1)$ denote a fixed parameter. Let $\mu_{\min}$ denote the minimal density of sates in $\mathcal{S}$. There is an iterative algorithm (Algorithm 5) with $c^{-2}\epsilon/4$ query quantization that spends $O(dn^{1+o(1)} \cdot \kappa)$ time in preprocessing and $O((n^{1+o(1)} + dn) \cdot \kappa)$ space, takes $O(\frac{\beta D_{\max}^2}{\epsilon^2(1-\gamma)^3\mu_{\min}^2})$ iterations and $O(dn^\rho \cdot \kappa + \mathcal{T}_Q)$ cost per iterations, start from a random point $\pi_\theta^0$ as initial point, and output policy $\pi_\theta^T$ that has average gap $\sqrt{\sum_{s \in \mathcal{S}} g_T(s)^2} < \epsilon$ holds with probability at least $1 - 1/\operatorname{poly}(n)$, where $g_T(s)$ is defined in Eq. (26) and $\kappa := d\log(ndD_X/(\lambda\delta))$.*