# OpenReview forum: "Breaking the Linear Iteration Cost Barrier for Some Well-known Conditional Gradient Methods Using MaxIP Data-structures"
_NeurIPS.cc/2021/Conference — NeurIPS 2021 Poster_

### Official Review · Reviewer_Tjo5 · 2021-07-07

**Rating:** 4
**Confidence:** 4

**Summary:**

This paper focuses on the iterative conditional gradient methods and proposes a way to improve the per-iteration cost of such methods without significantly improving the number of iterations needed for convergence. The paper presents a way to pose the computational bottleneck in each iteration as a maximum inner product search (MaxIP) problem and leverages existing LSH-based approximate MaxIP schemes to theoretically improve the per-iteration cost while also quantifying the convergence rate of such a conditional gradient method that introduces approximation in each iteration. Specifically, the conditional gradient method involved in the Frank-Wolfe scheme, the kernel herding scheme and the Policy gradient scheme are studied.


**Limitations And Societal Impact:**

This paper focuses on improving the efficiency of general optimization algorithms and hence has no negative societal impact.

The paper does not explicitly discuss any limitations even though the checklist points to section 3, which just states the problem setting and the main theoretical results without any discussion of limitations. Beyond the clarity issues, there are various limitations that I have highlighted in the **Main Review** which could have been listed as limitations.


**Main Review:**

## Positives:

The paper clear presents the research questions being focused on. These questions are of high importance since they address the efficiency of widely used optimization algorithms such as Frank-Wolfe and Policy Gradient. The authors leverage the existing theoretical guarantees for LSH-based MaxIP solutions to provide concrete computational gains from leveraging approximate MaxIP for direction search problem in these optimization algorithms.

The idea of posing the direction search in conditional gradient methods as a MaxIP problem by leveraging properties of a convex hull is a very interesting way of looking at this problem that allows us to leverage the efficient solutions for approximate MaxIP.

## Issues/concerns:

### Clarity and notational consistency/main vs. supplemental

The paper in its current version has a lot of notational issues and many parts are not clear or very incomplete in the main paper. Beyond proofs and such, it seems like various crucial information regarding problem setup and notation and algorithms have been placed in the supplement and it is hard to judge the significance of the contribution in the current form of the main paper. I needed the supplementary material to even understand the problem setup and the main contributions. Here are some specific examples:

- The theorem statements are not very well presented with some of the crucial parts of the theorems in the supplement or in other parts of the paper, which often leaves the theorems ill-stated.
- In my opinion, more important to have the Algorithms 1-3 in the main paper than the proof sketch.
- Table 1 says $n$ is the number of parameters while lines 145 and line 149 use $n$ to describe number of points in the set $A$ or $S$ over whose convex hull $\mathcal{B}(S)$ we are optimizing? This is confusing -- my assumption was that the number of parameters is $d$ since $S \subset \mathbb{R}^d$ and $w \in \mathcal{B}(S)$
- Theorem 3.2 appears mis-stated since $w^T \in \mathbb{R}^d$ while $\mu \in \mathbb{R}^k$ and hence the bound is meaningless without definitions of the l2-norm.
- In Theorem 3.3, what is $D_\max$ over action space for discrete action spaces -- in case of continuous action spaces, noone uses linear scan over action spaces ...
- In Def 4.5, again notional issues where the inner-product is being computed between vectors of different dimensionalities ($d$ vs $k$).
- The direction search described after line 128 appears to be over a convex set $S$ while the search defined in (2) is a direction search but does not ensure anything about the direction being in some convex set -- the mapping of notations and terms is not clear.
- The definition of "maximum diameters $D_x$ and $D_y$" are not clear and needs to be corrected/clarified. These quantities should be independent of specific $y$ or else the parity is not true.
- There is no $x,y \in \mathbb{R}^d$ used in line 306 -- we care about $s \in \mathbb{R}^k$ and $a \in \mathbb{R}^d$.


### Motivation for problem setup

Usually in an optimization problem, we have the definition of the convex set $\mathcal{B}$. In the problem setup for this paper, to leverage the MaxIP operation, we need to select the set $S$ of $n$ points such that $\mathcal{B} \subseteq \mathcal{B}(S)$. Once we have that set $S$, the results of this paper follow, but getting the set $S$ seems non-trivial in general convex optimization (or non-convex optimization) -- this could require $n = O(2^d)$, making the per-iteration cost quite prohibitive overall. For example, it would be good to understand what the size of $S$ needs to be to include a $\ell_p$-ball. Given this concern regarding the problem setup, here are some specific questions/comment:


- Line 131-132: why is the problem setup claimed to be more general while it seems to be less general. We might need precise details derivation of why "finite feasible $S$" is sufficient.
- Line 152-153 "In this setting ..": What is the motivation for studying this setup if we cannot get the "fast learning rates of Frank-Wolfe"?

The use of the MaxIP solver in the Policy Gradient algorithm appears very unclear. Upon parsing the supplement, I understand that we would need to generate LSH per state $s \in S \subset \mathbb{R}^d$. For continuous state spaces, this seems infeasible -- the preprocessing costs might be prohibitive -- it is not clear how the size of $S$ is considered in the preprocessing time. Is this a common setup in Policy Gradient algorithm where the size of $S$ is small and the set $\mathcal{C}(s)$ of possible actions for some state $s \in S$ is also a small convex set.

- The "Cost per iteration" discussion in lines 314-317 appear to be wrong given the description of the problem setup where $\mathcal{C}(s)$ would change with the state $s$, leading to different preprocessing for different states $s \sim \mu$. This needs to be clarified.
- Also, in the policy gradient part, it is not clear how the failure probability over all $|S|$-LSHs are handled given the state transitions can cause issues with independences. The size of set of states $S$ does not appear anywhere.

### Clarification regarding adaptive nature of the queries

Regarding section 4.5:
- This section addresses an important issue regarding the adaptive nature of the queries instead of the usual assumption of independence among queries.
- However, based on the discussion in the main paper, the independence aspect is not clear since the centers might not be independent as claimed -- the gradient method drives the selection of the query which in turn selects the hypercube in the lattice and the center of this hypercube is not random. So the selection of the hypercubes are not independent and hence the centers do not intuitively appear to be independent. Can this be clarified?


**Time Spent Reviewing:**

4

---

> ### Author Response · Authors · 2021-08-10
> **Thank you, please see the following clarifications.**
>
> We thank the reviewer for the comments!
>
> **Clarification on notations:**
>
> 1. We follow the writing style of theoretical paper, wherein the paper, we state the main theorem and the summary of techniques to prove the main theorem. Our supplementary material is self-contained to benefit the readers that require more details.
>
> 2. Same as point 1, we follow the theoretical paper writing style that states the main theorem and summary of techniques in the main paper due to the page limit. We provide a detailed algorithm structure in the self-contained supplementary.
>
> 3. As the reviewer pointed out in the Motivation for problem setup section, we focus on the optimization problem where we would like to minimize the function $f$ over the convex hull made up by $n$ number vectors in $\mathcal{R}^d$. Therefore, $n$ should be the parameter in the formulation and not $d$.
>
> 4. Thanks for catching the typo. It should be $w\in \mathcal{R}^k$ in Theorem 3.2. You can see we are using $\mathcal{R}^k$ in the proof.
>
> 5. Our analysis is on discrete settings. Continuous extensions will be future work and will likely require dealing with another level of approximation of continuous action space with discrete actions.
>
> 6. We will update the paper with correct notations.
>
> 7. In line 128, we state the general algorithm for Frank-Wolfe. We specify our problem setting in Section 3.1. Moreover, we provide a detailed formulation in Section C.1 of supplementary material.
>
> 8. Here $D_x$ and $D_y$ could be any large number that makes $\Vert x/D_x\Vert_2< 1$ and $\Vert y/D_y\Vert_2< 1$.
>
> 9. We will rectify x and y.
>
>
> **Clarification on the problem set up:**
>
> **Clarification on feasible set S:**
>
> 1. In this paper, we study the problem where we would like to optimize within the convex hull of a given finite vector set $S$. We state the generality of this setting as previous literature focuses on the optimization over $l_p$ balls, which are more regular. We don't need regularity. Just the size of $S$ should be under control. We can clarify our notion of generality if needed.
>
> 2. We state in lines 152-153 that this setup has various applications in practice. However, faster Frank-Wolfe algorithms for over $l_p$ balls could not be generalized to this setup. Therefore, we need to develop a new algorithm to reduce the cost per iteration instead.
>
> **Clarification on policy gradient:**
>
> 1. As mentioned, our setting is not continuous and the $C(s)$ for each state has $n$ number of actions. Discrete to continuous is another issue for future work. We believe we can even do continuous with some kind of epsilon net decomposition and hashing the “anchor points”. We will add more discussion on this topic in the final version of the paper.
>
> 2. We start with the situation that $C(s)$ is the same for every state. We will add more discussion on various $C(s)$ in the future. As the query of each state’s LSH is independent, we could union bound the failure probability.
>
> **Clarification on adaptive queries:**
>
> In the Frank-Wolfe algorithm, the gradients computed at every step are not statistically independent. Therefore, the queries containing gradients at each step are not independent. In this case, we apply adaptive query design for MaxIP data structures.
>
> We hope our response clarifies the reviewers’ doubts and confusion about our settings and results.

---

### Official Review · Reviewer_QnkL · 2021-07-13

**Rating:** 8
**Confidence:** 4

**Summary:**

The paper proposes a strategy to reduce the per-iteration cost of Frank-Wolfe, Herding and policy gradient methods. The algorithm is based on maximum inner product search (MIPS), equipped with proper projection and data transformations. These techniques transforms the problem into an ANN problem, where LSH is applied to achieve improved sublinear complexity. Theorems with rigorous statements on the improvement in complexity are provided on three algorithms, with clearly sketched proof to follow. Practical tricks using quantization and the utility-efficiency tradeoff is also discussed.

**Main Review:**

The paper is clearly presented and organized and is easy to follow. The theoretical results are solid and novel. The paper discusses three algorithms, which would be interesting to audiences from different communities. The theoretical contribution of this work is strong. The main result that applying LSH leads to reduced complexity but same convergence rate is also very interesting to me.

LSH is an useful tool that can be used in many aspects of computer science. I think it is a good paper presenting an novel and effective application of LSH to Frank-Wolfe type optimization methods. This could lead to more meaningful works in this direction.

Detailed comments:

1) I notice that the rates in Table 2, 3, 4 in the Appendix have two more constants $c$ and $\kappa$. They are determined by what factors? Also, what is $\rho$ commonly in practice?


**Time Spent Reviewing:**

2

---

> ### Author Response · Authors · 2021-08-10
> **Thank you for the comprehensive review, please see the following clarifications.**
>
> We thank the reviewer for the strong support of our work!
>
> The $c$ is a user set approximation parameter for approximate MaxIP in Definition A.4. $\kappa=\Theta(\log(T/\delta))$ , where $T$ is the number of iterations and $\delta$ is the failure probability. We repeat LSH data structures for $\kappa$ time to boost the success probability.
>
> Given the user set $c$ and $\tau$ in Definition A.4, $\rho=\frac{2(1-\tau)^2}{(1-c\tau)^2}-\frac{(1-\tau)^4}{(1-c\tau)^4}+o(1)$, where $o(1)=O(1/\sqrt{\log n})$.
> Let $c\in [0.5,1)$ and $\tau\in [0.5,1)$, we show that $\rho$ is decreasing as $\tau$ increases. Moreover, we could bound the $\rho$ as $\rho<1-\frac{(1-c)^2}{4}+O(1/\sqrt{\log n})$. We will add the statement along with the proof in the paper.

---

### Official Review · Reviewer_krev · 2021-07-16

**Rating:** 6
**Confidence:** 2

**Summary:**

The paper formally proves the correctness and efficiency of using Locality Sensitive Hashing to reduce the runtime-per-iteration of Conditional Gradient based algorithms when the feasible space is the convex hull of $n$ points. Owing to this convex hull, finding the conditional gradient update vector at every iteration involves solving a Maximum Inner Product (MaxIP) problem:
> Given a set $S$ of $n$ vectors (i.e. the convex hull) and a query vector $q$ (i.e. the gradient), find which vector in $S$ has maximum inner product with $q$.
The paper approximately solves the MaxIP problem by transforming it to an Approximate Nearest Neighbors problem, which is solved black-box by Locality Sensitive Hashing problems. The paper then gives formal guarantees on convergence, showing that this approximation lets each iteration runs asymptotically faster while the total number of iterations needed is asymptotically unchanged.

Guarantees are specifically given for Frank-Wolfe, Herding, and Policy Gradient algorithms. The policy gradient algorithm is constrained to a Markov Decision Process where the states and actions are real vectors, and there exists a finite set of actions.

**Limitations And Societal Impact:**

The authors do not seem to directly discuss the limitations of their work anywhere.

Societal Impact: Not Applicable

**Main Review:**

### Overall Impression

The paper is fine. It delivers on an interesting direction of improvement, but needs some clarity fixes.

Notably, this paper is also further from my area of expertise than I originally expected.

### Detailed Review

The paper also is ***littered*** with minor typographical and grammatical errors after the introduction. While this is not grounds for rejection, this was extraordinarily frustrating to read, and the authors should be sure to properly proofread their writing. After the introduction, I found about 8 paragraphs *without* at least one such typo. A full list of typos I found in the first 200 lines is at the end of this review.

The paper has a clear goal: theoretically analyze how LSH can speedup conditional gradient methods. The restricted setting of convex hulls of a finite set of vectors seems reasonable and interesting. The proofs seems simple and approachable. The guarantees as stated in Section 3.2 lack some clarity (details later), but are overall interesting and appear to be an improvement.

The significance of this paper is hard for me to comment on, as I am not particularly well versed in the Conditional Gradient literature. Taking the introduction as gospel, it seems publication-worthy to provide the first sublinear-in-$n$ runtime-per-iteration Frank Wolfe algorithm.

I have a couple clarity questions on the main results, as stated in Section 3.2:

- The role of $\rho$ is unclear:
    - Does the user set LSH parameter $\rho$ or is it defined by other parameters? I understood it to be user-set, but this is not clear to me. If it is not user set, then you should state it's rate, so we can better interpret the $O(dn^\rho)$ runtime. *i.e. Just how sublinear is this setting?*
        - [Line 159] The description in Table 1 makes $\rho$ sound defined by other parameters.
        - [Line 164] makes $\rho$ sound user-set.
        - [Line 788 of the appendix] makes $\rho$ sound defined by other parameters, and further that $\rho \approx 2\frac{(1-\tau)^2}{(1-c\tau)^2} + o(1)$, where $\tau$ depends on the vectors in $S$ and $c$ is the approximation quality in LSH. The dependence of $o(1)$ is also unclear here; does it depend on $n$ or $\tau$ or $c$? Is $\rho<1$ always, or just asymptotically?
    - If $\rho$ is user-set, the cost of making $\rho$ small should be stated. If the per-iteration-cost of your algorithm is $O(dn^\rho)$ for user-fixed parameter $\rho\in(0,1)$, why not make $\rho$ arbitrarily small? Glancing at citation [66], I believe the answer lies in both the space complexity and preprocessing time?

I have one last technical question:
- [Line 249] In Section 4.2, what is meant by $D_x$ being the diameter of vector $x$? Is this different from the norm of $x$? If so, where is this defined? If not, why not use the norm symbol instead?

---

### The long list of typos

Remember, this is just the first 200 lines of the 350 line paper.

1. [Line 78] Should be "**concludes** the paper"
1. [Line 92] Should be "any two **points**""
1. [Line 92] I recommend writing "uniformly at random from $H$", but it's not vital
1. [Line 96] Should be "first **shows** that"
1. [Lines 100, 102] Should be "directly" not "direct"
1. [Line 107] Should be "networks" not "network"
1. [Line 108] Confusion between singular vs. plural in the sentence starting with "In same task"
1. [Line 108] Should be "In the same task" not "In same task"
1. [Line 111] Should be "updates" not "update"
1. [Line 115] The sentence starting with "In bandit problem" is broken. I'm 100% sure what it's trying to say.
1. [Line 126] Replace the period after "maximum inner product" with a comma
1. [Line 128] Should be "updates" not "performs update"
1. [Line 138] Should be "iteration" not "iterations"
1. [Lines 149, 162, 164, 177, 188, 189, 190, 191] Should be "denote" not "denotes"
1. [Line 150] Should be one sentence with a comma instead of a period.
1. [Line 153] Should be "cannot" not "could not"
1. [Line 153-154] Broken sentence.
1. [Line 158] "iteration to be sublinear" not "iteration to sublinear"
1. [Line 159: Table 1 Description] Should be "denote" not "denotes" after both $T_g$ and $\gamma$. Should be "$T_Q$ denotes the time" not "$T_Q$ the time". Should be "convex" not "coonvex".
1. [Line 164] "be the convex hull of $S$" not "is the convex hull of $S$"
1. [Line 166] "with that takes" should be "that takes"
1. [Line 167] should be "as its initialization point, and" not "as initialization point,and"
1. [Line 170] "The Herding Algorithm" not "Herding algorithm"
1. [Line 179] "as its initialization point" not "as initialization point"
1. [Line 193] should be "iteration, starts from a random pint $\pi_\theta^0$ as its initial point, and outputs $\pi_\theta^T$ that has average gap"
1. [Line 198] "into a MaxIP" not "as a MaxIP"
1. [Line 199] "detailed proofs are" not "detailed proof is"

Counting repeated errors, that's about one typo every 5 or 6 lines, which is biased by the first 70 lines being typo-free. After section 1, it's about one typo every 3 or 4 lines. This typo rate continues through the end of the paper.

**Time Spent Reviewing:**

5

---

> ### Author Response · Authors · 2021-08-10
> **Thank you for the support, please see the following clarifications.**
>
> Thanks for your encouragement and suggestions, which will help us improve the paper! We have carefully thought through all your great questions and will add corresponding discussions to answer them in the updated paper. We provide details below:
>
>
> **1. The role of $\rho$.**
>
> The $\rho$ is determined by the two user set parameters: $c$ and $\tau$. $c$ and $\tau$ are the two parameters for approximate MaxIP in Definition A.4. Given the user set $c$ and $\tau$, $\rho=\frac{2(1-\tau)^2}{(1-c\tau)^2}-\frac{(1-\tau)^4}{(1-c\tau)^4}+o(1)$, where $o(1)=O(1/\sqrt{\log n})$.
>
>
> Let $c\in [0.5,1)$ and $\tau\in [0.5,1)$, we show that $\rho$ is decreasing as $\tau$ increases. Moreover, we could bound the $\rho$ as $\rho<1-\frac{(1-c)^2}{4}+O(1/\sqrt{\log n})$. Therefore, given large $n$, we have $\rho<1$. We will add the statement along with the proof in the paper.
>
>
> **2. The Definition of $D_x$ and $D_y$:**
>
> Here $D_x$ and $D_y$ could be any large number that makes $\Vert x/D_x\Vert_2< 1$ and $\Vert y/D_y\Vert_2< 1$. We should set them to be greater than the norm of the vector.
>
>
> List of Typos:
>
> We thank the reviewers for identifying the typos in the paper. We will revise the paper with the new edition to eliminate the typos.

---

### Official Review · Reviewer_UDJZ · 2021-07-16

**Rating:** 7
**Confidence:** 3

**Summary:**

The authors demonstrate a theoretical analysis for conditional gradient methods with sublinear per iteration complexity. Extension to Frank-Wolfe algorithm, Herding algorithm, and policy gradient methods are also provided with the same argument.

**Ethics Review Area:**

["I don’t know"]

**Limitations And Societal Impact:**

Yes.

**Main Review:**

The proposed analysis for CGM in terms of the per iteration time complexity is first type (as the authors claim) of such result that can be applied to related problems (e.g., Frank-Wolfe algorithm as shown in the paper). This is an interesting and also important result, which can help further help quantify the overall time complexity for related algorithms as most existing theoretical efforts were about the outer loop complexity.

The proposed analysis is based on transformation of direction search into a projected approximate maximum inner product search problem, with the asymptotic behavior of the same order in the iteration complexity towards convergence for the proposed sublinear Frank-Wolfe algorithm, which is an interesting approach and can be extended in the analysis of related problems.

The paper is relatively easy to follow and clearly presented. The presented proof sketch is mostly clear and seems correct, though I did not check all detailed analysis for the proof. Overall, I think the paper overall shows an interesting result.

**Time Spent Reviewing:**

5

---

> ### Author Response · Authors · 2021-08-10
> **Thank you for the comprehensive review.**
>
> We appreciate your concise and precise summarization of our work!

---

### Decision · Program_Chairs · 2021-09-27

**Decision:**

Accept (Poster)

**Comment:**

Overall, the majority of the reviewers thought that the combination of MaxIP data structures with conditional gradient methods for continues optimization, even though some of the main derivations are somewhat straightforward and not particularly difficult, is interesting enough to justify publication in NeurIPS, and may encourage further research along these line. I also agree with this view.